# SMOTE AND MIRRORS: EXPOSING PRIVACY LEAKAGE FROM SYNTHETIC MINORITY OVERSAMPLING

**Georgi Ganev**[1,2]    **Reza Nazari**[1]    **Rees Davison**[1]    **Amir Dizche**[1]    **Xinmin Wu**[1]
**Ralph Abbey**[1]    **Jorge Silva**[1]    **Emiliano De Cristofaro**[3]
[1]SAS    [2]UCL    [3]UC Riverside
georgi.ganev@sas.com

## ABSTRACT

The Synthetic Minority Over-sampling Technique (SMOTE) is one of the most widely used methods for addressing class imbalance and generating synthetic data. Despite its popularity, little attention has been paid to its privacy implications; yet, it is used in the wild in many privacy-sensitive applications. In this work, we conduct the first systematic study of privacy leakage in SMOTE: we begin by showing that prevailing evaluation practices, i.e., naive distinguishing and distance-to-closest-record metrics, completely fail to detect any leakage and that membership inference attacks (MIAs) can be instantiated with high accuracy. Then, by exploiting SMOTE's geometric properties, we build two novel attacks with very limited assumptions: `DistinSMOTE`, which perfectly distinguishes real from synthetic records in augmented datasets, and `ReconSMOTE`, which reconstructs real minority records from synthetic datasets with perfect precision and recall approaching one under realistic imbalance ratios. We also provide theoretical guarantees for both attacks. Experiments on eight standard imbalanced datasets confirm the practicality and effectiveness of these attacks. Overall, our work reveals that SMOTE is inherently non-private and disproportionately exposes minority records, highlighting the need to reconsider its use in privacy-sensitive applications and as a baseline for assessing the privacy of modern generative models.

## 1    INTRODUCTION

From rare disease diagnosis to fraud detection, machine learning tasks can be profoundly affected by severe class imbalance, where instances of interest – the minority class – are much rarer than the majority class (He & Garcia, 2009). Models often underperform under these conditions, exhibiting biases toward the majority and failing to capture the minority reliably (Chen et al., 2024a). One of the most influential and widely adopted approaches to address this is the Synthetic Minority Over-sampling Technique (SMOTE) (Chawla et al., 2002), which augments the imbalanced data by upsampling or generating synthetic samples of the underrepresented class through linear interpolation between minority records. Due to its simplicity and effectiveness, SMOTE continues to play a central role in real-world applications. To put things in context, the SMOTE paper has been cited nearly 40k times, Microsoft Azure offers built-in SMOTE components (Microsoft, 2024; 2025), and most MLaaS services support it (Google Cloud, 2025; AWS, 2025). Overall, SMOTE is primarily used in two contexts: 1) as a data augmentation technique for machine learning classifiers, and 2) as a synthetic data generation method to facilitate data sharing.

**Data Augmentation.** SMOTE was originally proposed as a pre-processing/upsampling technique to augment the real dataset, thus improving classifier performance, especially F1 score and recall, when trained on the augmented data. Practitioners rely on SMOTE in a wide range of medical applications, including cancer diagnosis (Fotouhi et al., 2019), heart-related diseases (Muntasir Nishat et al., 2022; El-Sofany et al., 2024), diabetes prediction (Ramezankhani et al., 2016; Alghamdi et al., 2017), genetic risk prediction (Kosolwattana et al., 2023), etc. Beyond medicine, SMOTE is widely applied in finance, particularly for credit-card fraud detection (Zhao & Bai, 2022; Khalid et al., 2024) and predicting customer churn (Peng et al., 2023; Ouf et al., 2024).

**Synthetic Data.** SMOTE has also gained traction as a method for generating synthetic tabular data. Often used as a baseline for more advanced models like GANs and VAE, SMOTE has been shown

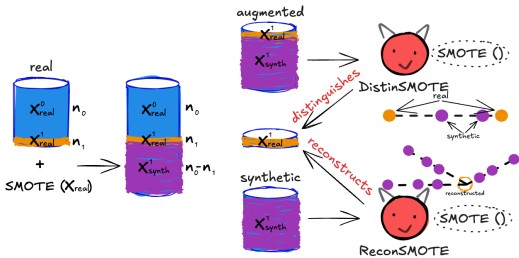

Figure 1: `DistinSMOTE` and `ReconSMOTE` attacks vs. augmented/synthetic data generated by SMOTE.

| Augmented Data | | |
|---|---|---|
| **Naive** | **MIA** | `DistinSMOTE` |
| $0.01 \pm 0.01$ | $0.68 \pm 0.07$ | $1.00 \pm 0.00$ |
| **Synthetic Data** | | |
| **Naive** | **MIA** | `ReconSMOTE` |
| $0.16 \pm 0.10$ | $0.93 \pm 0.02$ | $1.00 \pm 0.00$ |

Table 1: Performance of privacy attacks vs. SMOTE. Naive refers to the current privacy evaluation practices. MIAs are applied to SMOTE for the first time. `DistinSMOTE` and `ReconSMOTE` are our novel attacks.

to perform on par with, or even better than, generative approaches (Manousakas & Aydöre, 2023; Kindji et al., 2024). Moreover, its extensive use as a baseline has led to modern diffusion-based models (not explicitly designed with privacy in mind) to be characterized as privacy-preserving simply because they outperform SMOTE, a pattern repeatedly observed in top-tier machine learning publications (Kotelnikov et al., 2023; Zhang et al., 2024; Pang et al., 2024; Mueller et al., 2025). SMOTE is also applied for medical synthetic data (Kaabachi et al., 2025), and beyond machine learning research, it has been recognized as a promising technique for improving access to census data by public sector entities (ONS, 2019; ADR Wales, 2025).

**Roadmap.** Although not originally designed with privacy in mind, SMOTE is extensively used in sensitive public-facing applications that process personal data. However, its privacy risks have often been overlooked or significantly underestimated. In this paper, we fill this gap by studying whether, why, and how much privacy leakage occurs from using SMOTE either as a data augmentation method or as a standalone synthetic data generator.

We begin by showing that SMOTE appears to have no privacy leakage when evaluating it through the current practice of training a classifier to distinguish real from synthetic records in augmentation settings or that of measuring the distance from synthetic to real records (more precisely, the distance to closest record, or DCR (Zhao et al., 2021)). We refer to the former as naive distinguishing and the latter as naive metrics. We also instantiate—to our knowledge, for the first time—a Membership Inference Attack (MIA) (Shokri et al., 2017; Stadler et al., 2022) against SMOTE, showing that attackers can accurately infer whether a target record was part of the real training data.

Next, we propose two *novel* near-perfect privacy attacks with minimal and realistic assumptions: a Distinguishing (`DistinSMOTE`) and a Reconstruction attack (`ReconSMOTE`). Both only assume access to a single augmented or synthetic dataset and knowledge that SMOTE generated it (see Figure 1). By exploiting SMOTE's geometric properties, `DistinSMOTE` distinguishes real from synthetic records in augmentation settings, while the more ambitious `ReconSMOTE` reconstructs real minority records from synthetic data. We also provide a theoretical analysis for both attacks, showing they run at worst in $\mathcal{O}(n^2 d + n(kr)^2)$, where $n$, $d$, and $k$ denote, respectively, the number of input records, features, and SMOTE neighbors, and $r$ represents the data imbalance ratio. While quadratic in $n$, the complexity remains practical (especially with optimized search), with both attacks running within minutes on all datasets we experiment with.

`DistinSMOTE` achieves perfect precision and recall, while `ReconSMOTE` reaches perfect precision – which is more critical in privacy attacks (Carlini et al., 2022) – with recall increasing exponentially (with rate $\approx r/k$), reaching 1 under realistic parameter values ($k = 5$, $r \geq 20$).

Our experiments, summarized in Table 1, on eight standard imbalanced datasets demonstrate that:

- Naive distinguish (0.01 precision)/metrics (0.16 accuracy) completely underestimate risks.
- State-of-the-art MIAs achieve 0.68 AUC on augmented and 0.93 on synthetic data for 100 vulnerable targets, although being time-consuming. Also, sensitivity of targets increases when classifiers are trained on augmented vs. real data, yielding a 17% rise in MIA AUC.
- `DistinSMOTE` perfectly detects the real records in an augmented dataset.
- `ReconSMOTE` achieves perfect precision when reconstructing real minority records from a single synthetic dataset. While its average recall is 0.85, it reaches 1 for imbalance ratios of 20 or higher, consistent with our theoretical analysis.

**Implications.** Our findings provide further evidence that privacy cannot be treated as an afterthought when applying non-private techniques like SMOTE in sensitive settings. Its use not only risks exposing individual records but can also undermine trust in data-driven systems that rely on synthetic data. Overall, our work has the following real-world implications for researchers/practitioners:

1. SMOTE is fundamentally non-private: its interpolation process makes privacy leakage inherent, not a matter of flawed implementation.
2. Minority records are disproportionately at risk: the very samples SMOTE aims to amplify and make more representative are also the most exposed.
3. SMOTE and DCR are unreliable: evaluating SMOTE with privacy metrics like DCR gives a misleading assessment of its privacy and should not be used to validate other generative models.
4. Caution is critical: performance gains from oversampling must be weighed vs. privacy risks.

## 2 PRELIMINARIES

**Notation.** Let $D_{real} = (X, y)$ be a training dataset, where $X \subseteq \mathbb{R}^{n \times d}$ is the feature matrix consisting of $n$ samples with $d$-dimensional feature vectors and $y \in \{0, 1\}^n$ the corresponding binary labels. The dataset consists of $n_0$ majority and $n_1$ minority records, with imbalance ratio $r = \frac{n_0}{n_1} > 1$. In practice, $n_0 \gg n_1$, which makes learning directly from the minority class challenging.

**SMOTE** (Chawla et al., 2002) addresses class imbalance by generating synthetic minority samples; see Algorithm 1. To create a synthetic record, a random minority record from $D_{real}$ is selected, one of its $k$ nearest minority neighbors is chosen, and a new point is generated by interpolating along the line segment between them. Repeating this process yields $D_{syn}$ with $n_0 - n_1$ new samples, balancing the class distribution. The synthetic data can be used as a standalone synthetic dataset ($D_{syn}$) or to form an augmented dataset $D_{aug} = D_{real} \cup D_{syn}$, e.g., to improve classification performance.

---

**Algorithm 1** SMOTE (Chawla et al., 2002)

**Require:** Real dataset $D_{real}$
**Require:** Number of neighbors $k$
**Ensure:** Augmented data $D_{aug} = D_{real} \cup D_{syn}$, or Synthetic data $D_{syn}$
1: Filter minority $X^1_{real} \leftarrow \{X_{real}[i] | y_{real}[i]=1\}$
2: Compute $n_1 = |X^1_{real}|$, $n_0 = |D_{real}| - n_1$
3: **while** $|D_{syn}| < n_0 - n_1$ **do**
4:     Randomly pick $x_i \in X^1_{real}$
5:     Find $k$ nearest neighbors of $x_i$, $N(x_i)$
6:     Randomly choose $x_j \in N(x_i)$
7:     Sample $u \sim U(0, 1)$
8:     Generate $x_{syn} \leftarrow x_i + u(x_j - x_i)$
9:     Add $(x_{syn}, 1)$ to $D_{syn}$
10: **end while**

---

**Privacy Attacks.** Membership Inference Attacks (MIAs) (Shokri et al., 2017; Stadler et al., 2022) and Reconstruction Attacks (Dinur & Nissim, 2003; Annamalai et al., 2024a) are standard tools to empirically measure privacy leakage in ML. In MIAs, the adversary aims to infer whether a target record $(x_T, y_T)$ was part of the training dataset $D_{real}$. The attack can be framed as a repeated binary classification game: the adversary is given either a classifier (or a synthetic dataset) trained on $D_{real}$, or one trained on the neighboring $D'_{real} = D_{real} \setminus (x_T, y_T)$, and infers which dataset was used. To do so, the adversary typically exploits differences in model behavior – such as prediction confidences on the target, or statistical features extracted from synthetic data.

In a reconstruction attack, the adversary aims to recover any full real records (i.e., an untargeted attack) from access to a released model or synthetic data. These attacks often assume access to auxiliary information, such as public data, accurate statistics, or limited query access to $D_{real}$.

We also consider distinguishing attacks, which are somewhat related to MIAs but focus on whether a record comes from the population-level data distribution rather than from the specific dataset used to train the model. In the context of SMOTE, we use these attacks to distinguish unlabelled records in $D_{aug}$ as either real ($D_{real}$) or synthetic ($D_{syn}$), since $D_{real} \cap D_{syn} = \emptyset$.

## 3 RELATED WORK

As discussed in Section 1, SMOTE is widely used for data augmentation and synthetic data generation across various domains. Despite its popularity, prior work has focused primarily on its utility, while its privacy risks remain largely unexplored.

SMOTE has recently served as a baseline in several diffusion-based generative models (Kotelnikov et al., 2023; Zhang et al., 2024; Pang et al., 2024; Mueller et al., 2025), all published at top-tier machine learning venues. A common pattern across these studies is the "SMOTE + DCR" workflow: they rely on the Distance to Closest Record (DCR) (Zhao et al., 2021) as the primary privacy proxy, consistently reporting smaller DCR values for SMOTE and interpreting this as evidence that the newly proposed models are privacy-preserving. Kotelnikov et al. (2023) additionally employ the "full black-box" attack from (Chen et al., 2020), which still reduces to DCR as its core signal.

More recently, this view has been challenged by Sidorenko & Tiwald (2025), who show that some diffusion models (Kotelnikov et al., 2023; Mueller et al., 2025) actually achieve lower DCR values than SMOTE, thereby leaking more information about the training data. However, DCR itself has been shown to be an inadequate privacy metric – it consistently underestimates leakage (Houssiau et al., 2022; Annamalai et al., 2024b; Ganev & De Cristofaro, 2025) and does not correlate with leakage detected by MIAs (Yao et al., 2025). To the best of our knowledge, despite its prominent role as a baseline, SMOTE has not yet been systematically evaluated with state-of-the-art MIAs or any model-specific attacks. This leaves a critical gap in understanding SMOTE's true privacy risks and calls into question the validity of privacy claims across a recent line of generative-model research.

# 4 PRIVACY ATTACKS VS. SMOTE

In this section, we present our two novel privacy attacks that expose privacy leakage from SMOTE.

## 4.1 ADVERSARIAL MODEL

**Assumptions.** For both attacks, we assume an adversary with access to a *single* dataset generated by SMOTE ($D_{aug}$ for DistinSMOTE and $D_{syn}$ for ReconSMOTE). The adversary knows that the original SMOTE algorithm (Chawla et al., 2002) was applied (Algorithm 1) and is aware of its parameters, specifically, the number of neighbors $k$ and the real data imbalance ratio $r$ (in practice, these can be approximated from the released dataset). The adversary relies solely on the geometrical properties of SMOTE to achieve its goals – either distinguishing or reconstruction. Unlike prior privacy attacks, no further knowledge is required: e.g., the adversary does not need access to public/representative data, repeated inference or generation, the model parameters, numerous shadow models or a meta-classifier (as in MIAs (Stadler et al., 2022; Houssiau et al., 2022; Annamalai et al., 2024b)), or published (accurate) aggregate statistics (as in reconstruction (Dinur & Nissim, 2003; Dick et al., 2023)).

**Objectives.** For DistinSMOTE, the adversary aims to distinguish the real minority records from synthetic ones from observing $D_{aug}$, whereas for ReconSMOTE, to reconstruct them from $D_{syn}$. We focus on minority records from underrepresented regions of the feature space because they often correspond to the most vulnerable individuals. Such records carry the greatest privacy risks: they are easier to single out, more likely to be re-identified, and any disclosure disproportionately affects the individuals they represent (Kulynych et al., 2022; Stadler et al., 2022). Indeed, regulators, including the UK Information Commissioner's Office (ICO, 2022), have explicitly stressed the need to protect minority and outlier records.

We measure the attacks success using precision (the fraction of identified records that are truly real) and recall (the fraction of successfully identified real records), two standard metrics that together give a comprehensive view of performance. In privacy attacks, precision is especially critical, since even a handful of correctly identified records with high confidence can constitute a serious breach (Carlini et al., 2022). While secondary, capturing a large fraction of minority records is also important.

**Data Assumptions.** Our theoretical analysis of DistinSMOTE and ReconSMOTE relies on the following three realistic assumptions about the feature structure of the real minority records ($X^1_{real}$) and $k$:

*Assumption 1* (Real-valued features)*. All features in $X^1_{real}$ are real-valued.*

*Assumption 2* (Global non-collinearity)*. No three distinct feature vectors in $X^1_{real}$ are collinear.*

*Assumption 3* (Minimum $k$)*. The number of neighbors, $k \geq 3$.*

In other words, all features are continuous and no three points lie on the same line. These assumptions align naturally with (high-dimensional) continuous data, are non-restrictive in practice, and, crucially, are satisfied by all datasets in our main experiments (see Section 5). Moreover, assuming $k \geq 3$

is necessary to uniquely identify the intersection point of the lines formed by vectors connecting neighboring points, and is a standard, practical choice in typical SMOTE configurations (default is 5).

## 4.2 DISTINGUISHING ATTACK

In Algorithm 2, we outline the `DistinSMOTE` attack, which distinguishes *real* minority records from synthetic ones within an augmented dataset. The attack exploits the fact that, among any three collinear points, the middle one must be synthetic, since real points are non-collinear and SMOTE generates points strictly between them. The algorithm begins by searching from the convex hull of the minority records and iteratively explores neighbors inwards. When a collinear triplet is found, its midpoint is pruned from the candidate set with real points, and its neighbors are added to the queue for further inspection.

**Complexity.** The nearest-neighbor search is the main cost. With brute-force search ($\mathcal{O}(nd)$ per query), each of the $n$ records requires finding $kr$ neighbors ($\mathcal{O}(nd)$) and checking all neighbor pairs ($\mathcal{O}((kr)^2)$), yielding a worst-case complexity of $\mathcal{O}(n^2d + n(kr)^2)$. In practice, optimized methods (e.g., KD/ball trees) reduce search to $\mathcal{O}(\log n)$ for small $d$ ($d \leq 32$ in our main datasets). Also, since $k$ and $r$ are typically (small) constants, the effective complexity is much lower, and the algorithm completes in under three minutes on all main datasets.

**Accuracy Analysis.** We analyze `DistinSMOTE`, with the theorem below formalizing the theoretical correctness of its labeling rule, achieving perfect precision and recall.

**Theorem 1** (`DistinSMOTE` perfect precision & recall). *Under Assumptions 1–2, the labeling rule in the `DistinSMOTE` attack achieves perfect precision and recall (with probability 1).*

*Sketch Proof.* By SMOTE construction, each synthetic point lies strictly on the line segment between two real points ($x_{syn} = x_i + u(x_j - x_i)$). Under the global non-collinearity assumption, no three real points are collinear, so any line in $D_{aug}$ containing at least three points consists of exactly two real endpoints $x_i, x_j \in X^1_{real}$ and one or more synthetic interior points $x_{m_1}, x_{m_2}, \cdots \in X^1_{syn}$.
`DistinSMOTE` follows this labeling rule: it finds lines via local search (lines 9-11 of Algorithm 2) and marks interior points as synthetic and endpoints as real (lines 12-13), while any real point not on such a line (i.e., not used in interpolation) is also labeled real. The local search guarantees all points are visited efficiently, and the labeling rule ensures all synthetic points are removed and all real points preserved. This leads to perfect precision and recall (with probability 1, except for negligible numerical precision effects that do not occur in practice). □

---

**Algorithm 2** `DistinSMOTE`

**Require:** Augmented data $D_{aug}$
**Require:** Number of neighbors $k$, imbalance ratio $r$
**Ensure:** Detected real minority records $C^1$
1: Filter minority $X^1_{aug} \leftarrow \{X_{aug}[i]|y_{aug}[i]=1\}$
2: Initialize candidate set $C^1 \leftarrow X^1_{aug}$
3: Initialize queue $queue \leftarrow H(X^1_{aug})$ ▷ convex hull
4: Initialize visited set $V \leftarrow \emptyset$
5: **while** $queue \neq \emptyset$ **do**
6:   **for** record $x_i \in queue$ **do**
7:     **if** $x_i \notin V$ and $x_i \in C^1$ **then**
8:       Add $x_i$ to $V$
9:       Find $2 \cdot k \cdot r$ nearest neighbors of $x_i$, $N(x_i)$
10:       **for** pairs of neighbors $(x_j, x_k) \in N(x_i)$ **do**
11:         **if** $x_i, x_j, x_k$ are collinear **then**
12:           Identify middle point $x_m \in \{x_i, x_j, x_k\}$
13:           Remove $x_m$ from $C^1$; Add $x_m$ to $V$
14:           Add $N(x_m) \cap C^1$ to $queue$
15:         **end if**
16:       **end for**
17:     **end if**
18:   **end for**
19: **end while**
20: **return** $C^1$

---

## 4.3 RECONSTRUCTION ATTACK

Algorithm 3 presents the `ReconSMOTE` attack, which operates solely on synthetic data. The attack relies on two main intuitions: 1) synthetic records lie along line segments connecting real minority points, so these lines can be detected by finding three or more collinear samples, and 2) such lines intersect exactly at the original real points. The algorithm begins by iteratively defining lines, searching each point and two of its neighbors for collinear triplets, and then extending them with additional collinear neighbors. For each line, the mean of its points is stored as a midpoint, providing a compact representation of the line's location. Next, the algorithm examines pairs of midpoints to identify intersection points of the corresponding lines, which serve as candidate real records. Finally, we retain only intersections supported by at least three distinct lines, filtering out spurious candidates.

**Complexity.** The worst-case time complexity is very similar to `DistinSMOTE`, i.e., $\mathcal{O}(n^2 d + n(kr)^2)$. While there are two additional factors, namely, $\mathcal{O}(nkr)$ for checking neighbors after identifying a collinear triplet and $\mathcal{O}(n^2)$ for finding line intersections, these are dominated by existing terms and can be ignored. The practical complexity is much lower, and the attack runs in at most three minutes on all main datasets.

**Accuracy Analysis.** Next, we analyze `ReconSMOTE`; the following theorems give theoretical guarantees for its reconstruction rule, achieving perfect precision and a lower bound on expected recall.

**Theorem 2** (`ReconSMOTE` perfect precision). *Under Assumptions 1–3, the records reconstructed by* `ReconSMOTE` *are guaranteed to be real, i.e., the attack achieves perfect precision (with probability 1).*

*Sketch Proof.* By SMOTE construction, each synthetic point lies on a segment strictly between two real endpoints. Accordingly, every detected line in $X^1_{syn}$ (obtained from collinear sets identified by the local search in lines 8-17 of Algorithm 3) corresponds uniquely to a pair of real records. The local search correctly groups all synthetic points from that pair into a single collinear set without introducing spurious collinearities.

Under the global non-collinearity assumption, the only point lying on three or more such lines is their shared real endpoint. `ReconSMOTE` exploits this by intersecting detected lines (lines 23-25) and retaining only points supported by at least three distinct lines (line 27). Since synthetic points lie on exactly one SMOTE line, whereas real endpoints lie on three or more, any retained intersection must be a true real point. Hence, every reconstructed record is real, so precision is 1 (with probability 1). $\square$

For clarity and tractability, the next theorem uses a simplified formulation with directed edges and a Poisson approximation, giving a conservative bound (the *approximate* bound) that ignores overlapping neighbor relations in the SMOTE graph.

**Theorem 3** (`ReconSMOTE` expected recall (approximate)). *Let $\lambda = \frac{n_0 - n_1}{n_1 k}$. Under Assumptions 1–3 and using Poisson approximation (treating the number of synthetic points per segment as Poisson), the expected recall of* `ReconSMOTE` *satisfies:*

$$\mathbb{E}[\text{Recall}] \geq \qquad (1)$$

$$\max\left\{ 0, \ \frac{k\left(1 - e^{-\lambda}\left(1 + \lambda + \frac{\lambda^2}{2}\right)\right) - 2}{k - 2} \right\}.$$

*Sketch Proof.* At each generation step, SMOTE first selects a minority record $x_i$ uniformly from the $n_1$ available, and then one of its $k$ nearest neighbors $x_j$ uniformly. Thus, each of the $n_1 k$ possible minority-neighbor (directed) segments is chosen with probability $\frac{1}{n_1 k}$ at every step. Let $C_{ij}$ denote the number of synthetic points generated on segment $(x_i, x_j)$. Since each of the $n_0 - n_1$ synthetic points is assigned independently to a segment with probability $\frac{1}{n_1 k}$, the vector of all $C_{ij}$ follows a multinomial distribution with $\sum_{ij} C_{ij} = n_0 - n_1$, and each $C_{ij}$ is marginally distributed as $\text{Binom}(n_0 - n_1, \frac{1}{n_1 k})$. For analytic tractability, we approximate this by $\text{Poisson}(\lambda)$ with mean $\lambda = \frac{n_0 - n_1}{n_1 k}$. This is standard when $n_0 - n_1$ is large and $\frac{1}{n_1 k}$ is small, which holds in practice.

---

**Algorithm 3** `ReconSMOTE`

**Require:** Synthetic data $D_{syn}$
**Require:** Number of neighbors $k$, imbalance ratio $r$
**Ensure:** Reconstructed real minority records $R^1$
1 Filter minority $X^1_{syn} \leftarrow \{X_{syn}[i] \mid y_{syn}[i] = 1\}$
2 Initialize reconstructed set $R^1 \leftarrow \emptyset$ and line support map $S \leftarrow \emptyset$
3 Initialize set of lines $\mathcal{L} \leftarrow \emptyset$, midpoints $\mathcal{M} \leftarrow \emptyset$
4 Initialize visited set $V \leftarrow \emptyset$
5 **for** record $x_i \in X^1_{syn}$ **do**
6   **if** $x_i \notin V$ **then**
7     Add $x_i$ to $V$
8     Find $2 \cdot k \cdot r$ nearest neighbors of $x_i$, $N(x_i)$
9     **for** pairs of neighbors $(x_j, x_k) \in N(x_i)$ **do**
10      **if** $x_i, x_j, x_k$ are collinear **then**
11       Form initial line $(x_i, x_j, x_k)$; Add $x_j, x_k$ to $V$
12       **for** neighbor $x_n \in N(x_i) \setminus \{x_i, x_j, x_k\}$ **do**
13        **if** $x_n$ collinear with line $(x_i, x_j, x_k)$ **then**
14         Add $x_n$ to line $(x_i, x_j, x_k)$; Add $x_n$ to $V$
15        **end if**
16       **end for**
17       Add line to $\mathcal{L}$
18       Compute mean of line points and add to $\mathcal{M}$
19      **end if**
20     **end for**
21   **end if**
22 **end for**
23 **for** pairs of midpoints $(m_p, m_q) \in \mathcal{M}$ **do**
24   Compute intersection point $x^*$ of lines corresponding to $m_p$ and $m_q$
25   Add $x^*$ to $R^1$ and record support line indices $\{p, q\}$ in $S(x^*)$
26 **end for**
27 Filter points in $R^1$ with $|S(x^*)| \geq 3$
28 **return** $R^1$

---

A segment is reconstructed if $C_{ij} \geq 3$. The probability of this is $p_{\text{edge}} = \Pr\{\text{Poisson}(\lambda) \geq 3\} = 1 - e^{-\lambda}\left(1 + \lambda + \frac{\lambda^2}{2}\right)$. Now consider a fixed record $x_i$, and let $S_i$ denote the number of reconstructed

segments incident to it. Therefore, $\mathbb{E}[S_i] = k\,p_{\text{edge}}$. Moreover, by Assumption 3, once $S_i \geq 3$, the point $x_i$ is uniquely identifiable, since three non-collinear reconstructed segments suffice to triangulate its location. To lower bound $\Pr\{S_i \geq 3\}$, observe that $\mathbb{E}[S_i] = \mathbb{E}[S_i \mathbb{I}\{S_i \leq 2\}] + \mathbb{E}[S_i \mathbb{I}\{S_i \geq 3\}] \leq 2\,\Pr\{S_i \leq 2\} + k\,\Pr\{S_i \geq 3\}$ (since $S_i \leq k$). As $\Pr\{S_i \leq 2\} = 1 - \Pr\{S_i \geq 3\}$, this yields $\mathbb{E}[S_i] \leq 2 + (k-2)\Pr\{S_i \geq 3\}$, hence $\Pr\{S_i \geq 3\} \geq \frac{\mathbb{E}[S_i]-2}{k-2} = \frac{k\,p_{\text{edge}}-2}{k-2} := A_{id}$.

This is the probability that $x_i$ is identifiable. Since recall is the fraction of minority records that are identifiable, its expectation equals the average of these probabilities over all $n_1$ records. Because each $x_i$ is treated symmetrically in SMOTE and we look at directed segments, the average equals the bound derived above. Hence we obtain the stated lower bound on the expected recall. $\square$

**Remarks.** By rearranging Equation 1, we get $1 - \mathbb{E}[\text{Recall}] \leq \frac{k}{k-2}\,e^{-\lambda}(1 + \lambda + \frac{\lambda^2}{2})$, which in turn means $\mathbb{E}[\text{Recall}] \to 1$ as $\lambda \to \infty$, with a convergence rate exponential in $\lambda(= \frac{n_0 - n_1}{n_1 k} = \frac{r-1}{k})$.

In Appendix A, we provide a more detailed analysis and derive a tighter bound without the simplifications of Theorem 3; we call it the *exact* bound. Finally, in Appendix B, we visualize the differences between the bounds under various conditions.

## 5 EXPERIMENTAL EVALUATION

We now evaluate the effectiveness of our novel attacks, along with additional methods geared to assess privacy leakage in SMOTE, both as a data augmentation and synthetic data generation technique. Specifically, we consider: 1) current practices such as naive distinguish (via a classifier) and privacy metrics (i.e., DCR from synthetic to real records (Zhao et al., 2021)), 2) state-of-the-art Membership Inference Attacks (MIAs) (Shokri et al., 2017; Carlini et al., 2022), which to the best of our knowledge have not yet been applied against SMOTE, and 3) the `DistinSMOTE` and `ReconSMOTE` attacks. Overall results are summarized in Table 1.

**Datasets.** We conduct our main experiments on eight standard imbalanced datasets, each with a binary classification task, obtained from the imblearn library (Lemaitre et al., 2017) (originally from the UCI ML Repository) and used in prior work (Ding, 2011; Rosenblatt et al., 2025) These datasets vary significantly in size (336 to 11,183 records), dimensionality (6 to 32 features), imbalance ratios (8.6 to 130), and prediction task (target), as shown in Table 2.

| Dataset | Target | $r$ | $n$ | $d$ |
|---|---|---|---|---|
| ecoli | imU | 8.6 | 336 | 7 |
| abalone | 7 | 9.7 | 4,177 | 10 |
| car_eval_34 | vgood | 12 | 1,728 | 21 |
| solar_flare_m0 | M-0 | 19 | 1,389 | 32 |
| car_eval_4 | vgood | 26 | 1,728 | 21 |
| yeast_me2 | ME2 | 28 | 1,484 | 8 |
| mammography | minority | 42 | 11,183 | 6 |
| abalone_19 | 19 | 130 | 4,177 | 10 |

Table 2: Main datasets overview, where $r$ denotes the imbalance ratio ($n_0/n_1$), $n$ the number of records, and $d$ the number of features.

**Implementations.** We use the standard imblearn (Lemaitre et al., 2017) implementation of SMOTE and sklearn (Pedregosa et al., 2011) for classifiers. Both `DistinSMOTE` and `ReconSMOTE` are highly efficient, running in under three minutes on any dataset from Table 2 on an Apple M4 MacBook with 24GB RAM. The naive methods are similarly fast, while the MIAs take up to 30 minutes per dataset. We will release the source code for our attacks along with the final version of the paper.

### 5.1 AUGMENTED DATA

We compare the three approaches on augmented data, with results for all datasets shown in Table 3.

**Naive Distinguish** is a popular but arguably misguided approach for telling apart real and synthetic records by training a classifier (Snoke et al., 2018; El Emam et al., 2022; Qian et al., 2023; DataCebo, 2025). Half of the real and half of the synthetic data are used to train a Random Forest classifier, with testing performed on the remaining data. For each dataset, we run 5 independent SMOTE generations and train 5 classifiers per run, reporting averaged results. The method severely underestimates privacy risk (see the two leftmost columns in Table 3; precision and recall $\approx 0$) as it is capable of capturing only distributional differences, not record-level leakage.

**MIA.** Next, we evaluate MIAs (Shokri et al., 2017; Carlini et al., 2022) using the repeated classification game from Section 2. For a given target record, we train 200 classifiers (a multi-layer perceptron

| Dataset | $r$ | Naive distinguish $D_{aug}$ | | MIA $D_{real}$ | $D_{aug}$ | DistinSMOTE $D_{aug}$ | |
|---|---|---|---|---|---|---|---|
| | | (Precision) | (Recall) | (AUC) | (AUC) | (Precision) | (Recall) |
| ecoli | 8.6 | $0.00 \pm 0.00$ | $0.00 \pm 0.00$ | $0.50 \pm 0.04$ | $0.50 \pm 0.05$ | $1.00 \pm 0.00$ | $1.00 \pm 0.00$ |
| abalone | 9.7 | $0.03 \pm 0.03$ | $0.00 \pm 0.01$ | $0.57 \pm 0.03$ | $0.58 \pm 0.04$ | $1.00 \pm 0.00$ | $1.00 \pm 0.00$ |
| car_eval_34 | 12 | $0.01 \pm 0.02$ | $0.01 \pm 0.01$ | $0.60 \pm 0.03$ | $0.73 \pm 0.08$ | $1.00 \pm 0.00$ | $1.00 \pm 0.00$ |
| solar_flare_m0 | 19 | $0.01 \pm 0.03$ | $0.00 \pm 0.01$ | $0.79 \pm 0.03$ | $0.97 \pm 0.03$ | $1.00 \pm 0.00$ | $1.00 \pm 0.00$ |
| car_eval_4 | 26 | $0.00 \pm 0.00$ | $0.00 \pm 0.00$ | $0.59 \pm 0.03$ | $0.75 \pm 0.10$ | $1.00 \pm 0.00$ | $1.00 \pm 0.00$ |
| yeast_me2 | 28 | $0.00 \pm 0.00$ | $0.00 \pm 0.00$ | $0.51 \pm 0.04$ | $0.57 \pm 0.09$ | $1.00 \pm 0.00$ | $1.00 \pm 0.00$ |
| mammography | 42 | $0.01 \pm 0.02$ | $0.00 \pm 0.00$ | $0.54 \pm 0.03$ | $0.56 \pm 0.04$ | $1.00 \pm 0.01$ | $1.00 \pm 0.00$ |
| abalone_19 | 130 | $0.00 \pm 0.00$ | $0.00 \pm 0.00$ | $0.58 \pm 0.05$ | $0.80 \pm 0.12$ | $0.99 \pm 0.02$ | $1.00 \pm 0.00$ |
| average | | $0.01 \pm 0.01$ | $0.00 \pm 0.00$ | $0.58 \pm 0.03$ | $0.68 \pm 0.07$ | $1.00 \pm 0.00$ | $1.00 \pm 0.00$ |

Table 3: Privacy attacks vs. augmented data.

| Dataset | $r$ | Naive metrics (Accuracy) | MIA (AUC) | ReconSMOTE (Precision) | (Recall) |
|---|---|---|---|---|---|
| ecoli | 8.6 | $0.19 \pm 0.15$ | $0.93 \pm 0.05$ | $1.00 \pm 0.00$ | $0.43 \pm 0.02$ |
| abalone | 9.7 | $0.21 \pm 0.17$ | $0.65 \pm 0.07$ | $1.00 \pm 0.00$ | $0.62 \pm 0.01$ |
| car_eval_34 | 12 | $0.00 \pm 0.00$ | $0.97 \pm 0.01$ | $1.00 \pm 0.00$ | $0.83 \pm 0.03$ |
| solar_flare_m0 | 19 | $0.03 \pm 0.06$ | $1.00 \pm 0.00$ | $1.00 \pm 0.00$ | $0.95 \pm 0.02$ |
| car_eval_4 | 26 | $0.01 \pm 0.04$ | $1.00 \pm 0.00$ | $1.00 \pm 0.00$ | $1.00 \pm 0.00$ |
| yeast_me2 | 28 | $0.20 \pm 0.12$ | $0.99 \pm 0.01$ | $1.00 \pm 0.00$ | $1.00 \pm 0.00$ |
| mammography | 42 | $0.25 \pm 0.15$ | $0.91 \pm 0.04$ | $1.00 \pm 0.00$ | $1.00 \pm 0.00$ |
| abalone_19 | 130 | $0.37 \pm 0.14$ | $1.00 \pm 0.00$ | $1.00 \pm 0.00$ | $1.00 \pm 0.00$ |
| average | | $0.16 \pm 0.10$ | $0.93 \pm 0.02$ | $1.00 \pm 0.00$ | $0.85 \pm 0.01$ |

Table 4: Privacy attacks vs. synthetic data.

with two hidden layers) on augmented datasets generated via SMOTE: half of the training datasets include the target record, and half exclude it. We then use the classifiers' predictions on the target to simulate an adversary's confidence in distinguishing membership, and calculate AUC. Following prior work (Ye et al., 2024; Guépin et al., 2024), we train target-specific attacks in a leave-one-out setting, which provides a more accurate estimate of privacy leakage. This procedure is repeated for 100 randomly selected targets (or all minority records), and we report the average. Overall, this requires training roughly 20k SMOTE models and classifiers per dataset.

Looking at Table 3 (fourth column), the average AUC is 0.68, with half of the datasets exceeding 0.7, which indicates substantial privacy leakage. The lowest scores appear in datasets with the smallest imbalance (ecoli and abalone), where the proportion of synthetic data is relatively low. Mammography also shows a low score, likely because its large number of records reduces the influence of any single individual. These results are therefore not entirely surprising.

We also conduct an additional MIA experiment, training classifiers solely on the real data, to test the intuition that SMOTE enhances the sensitivity of minority records in the augmented data, as they directly contribute to generating synthetic samples. As expected, targets become more vulnerable when augmentation is applied – average AUC increases by 17% (comparing the third and fourth columns in Table 3). Larger imbalance further amplifies this effect. While similar intuitions have been noted previously (Rosenblatt et al., 2025), they were not supported by empirical evidence.

**DistinSMOTE.** Finally, we run DistinSMOTE on 25 SMOTE generations and report average precision/recall (two rightmost columns in Table 3). As expected from our analysis, we achieve perfect results across all datasets and imbalance levels. This shows that merely knowing SMOTE was used for augmentation is enough for an adversary to perfectly identify real records with minimal effort.

## 5.2 SYNTHETIC DATA

Next, we evaluate all attacks on synthetic data; see Table 4.

**Naive Metrics.** A widely used approach for evaluating privacy in synthetic data is the Distance to Closest Record (DCR) (Zhao et al., 2021), which measures the average distance between synthetic and real records. DCR has been commonly applied to SMOTE and modern diffusion models (Kotelnikov et al., 2023; Zhang et al., 2024; Pang et al., 2024; Mueller et al., 2025), but its interpretation is

limited – an average distance alone provides little insight into privacy risks. To address this, we use a linkability attack (Giomi et al., 2022), which builds on DCR and reports the accuracy with which an adversary could link two partial feature sets of a real record using synthetic data. For each dataset, we train 5 SMOTE models and evaluate linkability 5 times with varying feature subsets.

The results are unstable (see the leftmost column in Table 4): scores differ from zero only for low-dimensional settings ($d \leq 10$), while higher-dimensional datasets yield large variances that render DCR unreliable. This is expected, as DCR treats all features equally and is known to be an inadequate privacy measure (Annamalai et al., 2024b; Ganev & De Cristofaro, 2025; Yao et al., 2025).

**MIA.** We evaluate MIAs on synthetic data using the repeated classification game (similar to Section 5.1). We rely on the GroundHog attack (Stadler et al., 2022), one of the most popular MIAs for synthetic tabular data. GroundHog extracts statistical features from generated datasets – such as column-wise minimum, mean, median, maximum, and pairwise correlations – and uses them to train a meta-classifier, which is then applied to unlabeled real and synthetic feature sets. To generate training features, we train 400 SMOTE models for in/out training features and another 200 SMOTE models for in/out testing features. Repeating this for 100 targets yields about 60k models per dataset.

As shown in Table 4 (second column), this results in substantial privacy leakage: AUC exceeds 0.9 in all but one dataset. The exception is abalone, which has the second-lowest imbalance ratio and the second-largest number of records, potentially leading to lower sensitivity. When imbalance increases (abalone_19), the MIA AUC rises to 1. Overall, these results demonstrate that SMOTE-generated data is highly susceptible to MIAs, even beyond trivial cases where data domain characteristics mainly drive leakage (Ganev et al., 2025a;b).

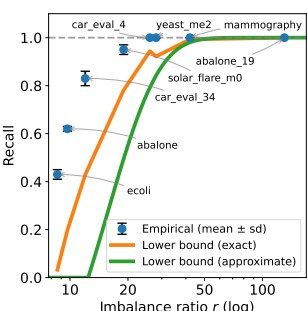

Figure 2: ReconSMOTE recall and lower bounds (per dataset).

**ReconSMOTE.** Next, we apply ReconSMOTE on 25 SMOTE generations per dataset, reporting average precision/recall (last two columns in Table 4). The attack achieves perfect precision on all datasets, which, as motivated in Section 4, is the most critical metric for reconstruction. Recall is also very high (see Figure 2), with an average of 0.85. It increases quickly with class imbalance, reaching 1 when $r \geq 20$. The recall values (per dataset) are in line with the expected approximate/exact bounds predicted by Theorem 3 and 4.

To further validate the expected bounds at finer granularity, we vary the imbalance ratio $\{5, 10, 20, 25, 50, 75, 100\}$ across all datasets and plot the average performance in Figure 3. As expected, recall increases exponentially with $r$ (for fixed $k$), reaching 1 around imbalance 20. Overall, these findings highlight the risks of relying on SMOTE for synthetic data generation: in realistic settings, an adversary can reconstruct all real records with perfect confidence.

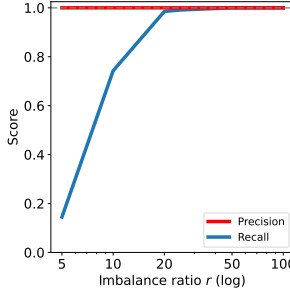

Figure 3: ReconSMOTE performance w/ varying $r$ (all datasets).

### 5.3 TAKE-AWAYS

We show that MIAs achieve high AUC across numerous targets vs. SMOTE: 0.68 against classifiers trained on augmented data and 0.93 against synthetic data. Moreover, the sensitivity of minority records increases by an average of 17% when classifiers are trained on augmented rather than original training data. Finally, our attacks, DistinSMOTE and ReconSMOTE, are able to i) distinguish real minority records from synthetic ones in augmented data, and ii) reconstruct real minority records from synthetic data with minimal assumptions and near-perfect accuracy.

## 6 DistinSMOTE AND ReconSMOTE WITH RELAXED ASSUMPTIONS

In this section, we test the robustness of our attacks, DistinSMOTE and ReconSMOTE, on augmented/synthetic data while relaxing our assumptions one at a time, e.g., using high-dimensional data, mixed-type data, and $k = 2$; results are shown in Table 5, 6, and 7 in Appendix C. Additionally, in Appendix C, we evaluate our attacks on perturbed SMOTE datasets (i.e., linear interpolation with added random noise) and provide a heuristic for running the attacks without knowledge of $k$ and $r$.

**High-Dimensional Data.** First, we test how the attacks scale to two high-dimensional datasets – up to 96,690 records and 50 features (see Table 5 and Appendix C for results and details about the higgs and miniboone datasets, both with $r = 25$). Both `DistinSMOTE` and `ReconSMOTE` achieve perfect precision and recall and, as analyzed in Section 4, scale well with $n$ and $d$, taking no more than 8 minutes per dataset, which is highly practical.

**Mixed-Type Data.** Next, we show that our attacks also apply to mixed-type data, relaxing Assumption 1; see Table 6 and Appendix C for details on the cardio and churn datasets, each containing an equal mix of numerical/categorical features, and with $r = 25$. To generate data, we use SMOTE-NC (Chawla et al., 2002) – introduced in the original SMOTE paper and designed for mixed data – via the standard imblearn implementation (Lemaitre et al., 2017). For the attacks, we simply ignore categorical features and operate on the continuous ones. As before, we obtain perfect precision and very high recall on both datasets. This approach can also be applied to one-hot encoded data.

**SMOTE with $k = 2$**. Finally, we evaluate the attacks on the eight main datasets using SMOTE with $k = 2$, relaxing Assumption 3 (see Table 7 in Appendix C). The performance of `DistinSMOTE` remains unaffected, achieving perfect precision and recall, as expected. As for `ReconSMOTE`, average recall drops to 0.52 (a 39% decrease compared to SMOTE with $k = 5$ in Table 4) because each real record participates in fewer lines, making it harder to reach the support threshold; only records that serve as neighbors to other records beyond their two closest neighbors can be successfully reconstructed. Precision remains perfect as all reconstructed points are still accurate. Nevertheless, reconstructing half the real minority records with perfect confidence is a serious privacy breach.

## 7 CONCLUSION

Our work highlights the fundamental privacy limitations of SMOTE (Chawla et al., 2002), one of the most widely adopted techniques for improved learning on imbalanced data. The effectiveness of our novel, near assumption-free attacks (`DistinSMOTE` and `ReconSMOTE`), demonstrates that real minority records – precisely the ones SMOTE aims to better represent – are exposed to significant, previously underestimated privacy risk. Importantly, this also shows that using SMOTE as a baseline with DCR to evaluate privacy is unreliable and can provide a false sense of security. Nonetheless, SMOTE remains an effective and easy-to-use technique in non-privacy-sensitive applications where utility is the primary concern. We are confident our findings will be valuable to researchers and practitioners deploying solutions that process or release sensitive data, motivate them to avoid SMOTE as a privacy benchmark, and encourage them to adopt more robust privacy-preserving techniques.

**Limitations and Future Work.** Our attacks currently operate on continuous data and are primarily tested on the original SMOTE implementation. While certain numerical instabilities/edge cases are theoretically possible (e.g., a synthetic point appearing collinear with two unrelated real points), their probability is effectively zero in high-dimensional datasets with high numerical precision, and we did not observe any such case in our experiments. Additionally, our findings generalize to many SMOTE variants – such as BorderlineSMOTE (Han et al., 2005), ADASYN (He et al., 2008), SVMSMOTE (Nguyen et al., 2009), and cluster/hybrid-based methods (Douzas et al., 2018) – as they all rely on line-segment interpolation to generate synthetic samples. In contrast, our attacks are unlikely to be successful against variants like G-SMOTE (Douzas & Bacao, 2019) and GI-SMOTE (Chen et al., 2024b), which generate synthetic points within regions rather than strictly along lines. Nevertheless, these variants are not inherently privacy-preserving and are still likely to remain vulnerable to MIAs. Extending our attacks and developing robust defenses for such methods is a promising direction for future work.

Privacy-preserving variants of SMOTE have also been proposed under the framework of Differential Privacy (Dwork et al., 2006; 2014), including DP-SMOTE (Lut, 2022), which adds noise when estimating point distributions/nearest neighbors, and SMOTE-DP (Zhou et al., 2025), which combines SMOTE with a DP generative model. However, SMOTE-DP largely ignores SMOTE's increased sensitivity of minority records (Lau & Passerat-Palmbach, 2021; Lut, 2022; Rosenblatt et al., 2025), a gap we confirm empirically (see Section 5.1). As none of these approaches provides open-source implementations, we leave evaluating their effectiveness to future work.

**Ethics Statement.** Our work does not involve attacking live systems or private datasets. Our goal is to demonstrate the importance of emphasizing privacy considerations and relying on established notions of privacy when processing sensitive, imbalanced data in critical domains.

We have only used Large Language Models (LLMs) to aid or polish writing. We performed all literature review, research ideation, and theoretical derivations.

**Reproducibility Statement.** We make considerable efforts to make our work reproducible. We clearly state all assumptions throughout the paper, provide detailed references and step-by-step explanations for accessing and preparing the datasets and privacy attacks used in our evaluation, and include pseudocode for our new attacks. Last, we release the source for our attacks and experiments: `https://github.com/sascommunities/smote-mirrors`.

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

## A  TIGHTER LOWER BOUND ON ReconSMOTE RECALL

In this section, we derive a tighter lower bound on the recall of ReconSMOTE by relaxing two of the assumptions in Section 4.3, namely the Poisson approximation and the one-directional counting of $C_{ij}$. Specifically, we use the exact Binomial distributions and count $C_{ij}$ in both directions to obtain more accurate values.

Recall that at each step of the SMOTE algorithm, we choose an $x_i \in X_{real}^1$ uniformly at random and then independently select one of its $k$ nearest neighbors uniformly at random. To capture this structure, we represent the minority data by a KNN graph $G = (X_{\text{real}}^1, E)$, where edges $E$ represent the neighboring relations among the minority samples. As before, we use $N(x_i)$ to denote the $k$ nearest points to $x_i$ from the minority set. For each $x_i \in X_{real}^1$, we add an edge $E_{i \to j}$ whenever $x_j \in N(x_i)$. Each synthetic data generated by SMOTE is associated with exactly one edge. Let $\alpha$ denote the probability that a nearest-neighbor relation is *mutual*; i.e., the probability that if $x_j \in N(x_i)$ then also $x_i \in N(x_j)$. In this case, a synthetic point lies on $E_{i \to j}$ if it was generated along $E_{i \to j}$ or along $E_{j \to i}$. If $\alpha = 1$, then all nearest-neighbor edges are mutual, and $\alpha = 0$ corresponds to a completely one-sided nearest-neighbor graph, for which we usually refer to those edges as *exclusive*.

Recall that an edge is *reconstructed* if at least three synthetic records lie on its segment, and a real $x_i$ is *identifiable* if there exist three reconstructed edges incident to $x_i$. We denote the number of

synthetic data points generated between $x_i$ and $x_j$ by $C_{ij}$. For generating each synthetic point, SMOTE performs these selections independently, assigning every new sample to exactly one of the $n_1 k$ directed edges with equal probability $1/(n_1 k)$. This independence is not an additional assumption – it follows directly from the random sampling mechanism of SMOTE. Consequently, the vector of all $C_{ij}$ follows a multinomial distribution with $\sum_{ij} C_{ij} = n_0 - n_1$, and each component $C_{ij}$ is marginally $\mathrm{Binom}(n_0 - n_1, \frac{1}{n_1 k})$.

Because some directed edges in the SMOTE KNN graph represent mutual neighbor relationships, certain edges overlap. We address this by distinguishing between one-way and mutual edges. Let $B_{ij} := \mathbb{I}\{x_i \in N(x_j) \text{ and } x_j \in N(x_i)\}$ (*mutuality indicator*) with $\Pr\{B_{ij} = 1\} = \alpha$. Then

$$C_{ij} \mid B_{ij} = 0 \sim \mathrm{Binom}\left(n_0 - n_1, \frac{1}{n_1 k}\right), \qquad C_{ij} \mid B_{ij} = 1 \sim \mathrm{Binom}\left(n_0 - n_1, \frac{2}{n_1 k}\right).$$

Consider the following assumption regarding the structure of neighboring relations around each real minority record.

**Assumption 4** (Local non-degeneracy). *For any $x_i \in X_{real}^1$, all edges $\{E_{i \to j} : x_j \in N(x_i)\}$ have pairwise distinct directions.*

In the analysis of this section, we use Assumption 4 in place of Assumption 2 (global non-collinearity), as it is a weaker, localized condition sufficient for establishing the lower bound on reconstruction recall. In particular, if Assumption 2 holds, then Assumption 4 automatically follows. Under Assumption 4, the intersection of any three reconstructed edges incident to $x_i$ uniquely identifies $x_i$.

**Lemma 1** (Reconstructed edge probability). *For any edge of $G$,*

$$\Pr\{C_{ij} \geq 3\} = (1-\alpha)\Pr\{\mathrm{Binom}(n_0 - n_1, \tfrac{1}{n_1 k}) \geq 3\} + \alpha \Pr\{\mathrm{Binom}(n_0 - n_1, \tfrac{2}{n_1 k}) \geq 3\} =: p_{\mathrm{edge}}(\alpha).$$

*Sketch Proof.* Condition on $B_{ij}$ and compute the average. If $B_{ij} = 0$, then only one direction contributes to the count; if $B_{ij} = 1$, both directions do. $\qquad\square$

**Lemma 2** (Lower-bound on per-node identifiability). *Fix $x_i \in X_{real}^1$ and its $k$ outgoing directed edges $\{E_{i \to j} : x_j \in N(x_i)\}$. Then, we have*

$$\Pr\{x_i \text{ identifiable}\} \geq \max\left\{0, \frac{k\, p_{\mathrm{edge}}(\alpha) - 2}{k - 2}\right\} =: L_{id}, \qquad (2)$$

*and this lower bound is tight.*

*Sketch Proof.* Declare an edge reconstructed if $C_{ij} \geq 3$ and set $E_{i \to j}^{\mathrm{rec}} := \mathbb{I}\{C_{ij} \geq 3\}$. Let $S_i = \sum_{j=1}^k E_{i \to j}^{\mathrm{rec}}$. From Lemma 1, each edge has marginal $\Pr\{E_{i \to j}^{\mathrm{rec}} = 1\} = p_{\mathrm{edge}}(\alpha)$, so $\mathbb{E}[S_i] = k\, p_{\mathrm{edge}}(\alpha)$. Then, we have

$$\begin{aligned}
\mathbb{E}[S_i] &= \mathbb{E}\big[S_i \,\mathbb{I}\{S_i \leq 2\}\big] + \mathbb{E}\big[S_i \,\mathbb{I}\{S_i \geq 3\}\big] \\
&\leq \mathbb{E}\big[2\,\mathbb{I}\{S_i \leq 2\}\big] + \mathbb{E}\big[k\,\mathbb{I}\{S_i \geq 3\}\big] \qquad \text{(since } S_i \leq k \text{ a.s.)} \\
&= 2\Pr\{S_i \leq 2\} + k\,\Pr\{S_i \geq 3\}.
\end{aligned}$$

Since $\Pr\{S_i \leq 2\} = 1 - \Pr\{S_i \geq 3\}$, we obtain

$$\mathbb{E}[S_i] \leq 2 + (k - 2)\Pr\{S_i \geq 3\},$$

hence

$$\Pr\{S_i \geq 3\} \geq \frac{\mathbb{E}[S_i] - 2}{k - 2} = \frac{k\, p_{\mathrm{edge}}(\alpha) - 2}{k - 2}.$$

Truncating at 0 accommodates the trivial case $k\, p_{\mathrm{edge}}(\alpha) \leq 2$.

Moreover, the bound cannot be improved using only the edge-wise success probabilities. Consider constructing $(E_{i \to j}^{\mathrm{rec}})_{j=1}^k$ so that $S_i = \sum_{j=1}^k E_{i \to j}^{\mathrm{rec}}$ takes values only in $\{2, k\}$. Choose the mixture weights so that $\mathbb{E}[S_i] = k\, p_{\mathrm{edge}}(\alpha)$. In this case, the inequality holds with equality, meaning that the lower bound is tight. $\qquad\square$

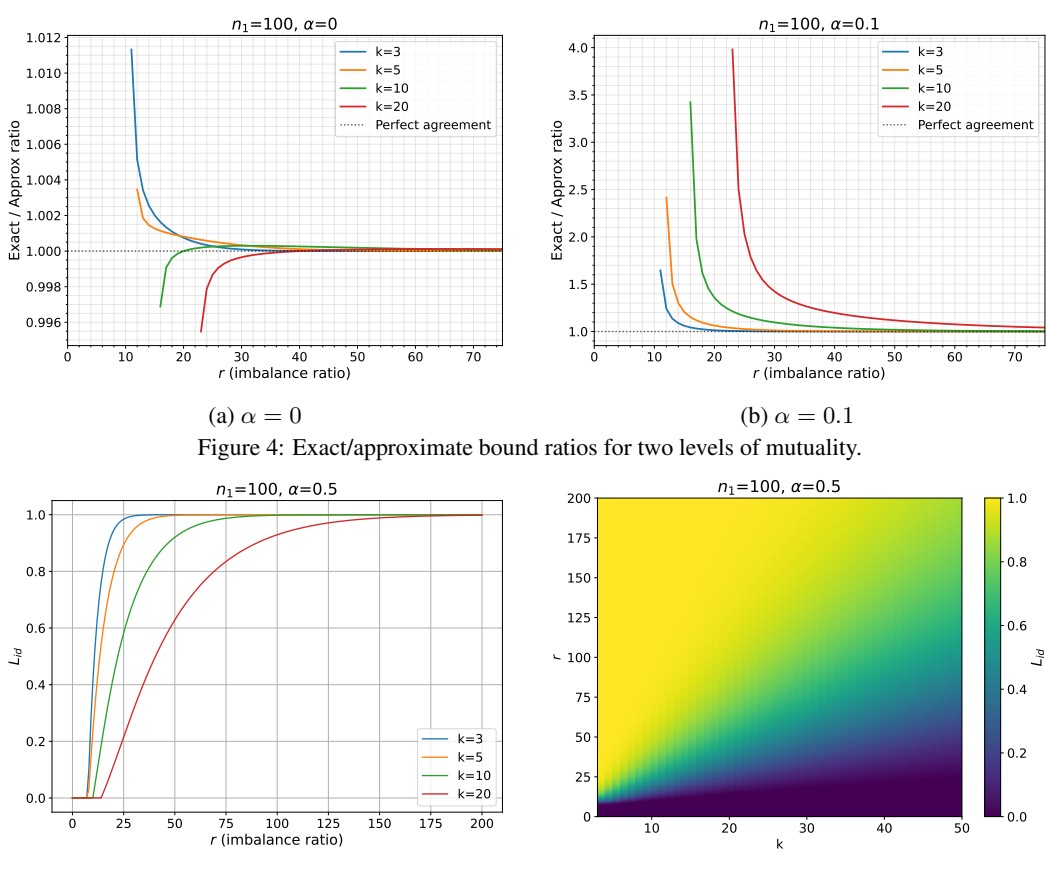

(a) $\alpha = 0$        (b) $\alpha = 0.1$

Figure 4: Exact/approximate bound ratios for two levels of mutuality.

(a) Lower bound $L_{id}$ vs. $r$       (b) Heatmap of $L_{id}$ over $(r, k)$

Figure 5: Comparison of two visualizations of the exact bound.

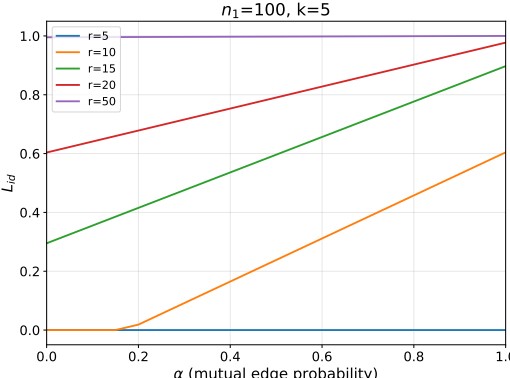

Figure 6: Exact bound as a function of $\alpha$ for different imbalance ratios $r$.

**Theorem 4** (`ReconSMOTE` expected recall (exact)). *Under Assumptions 1, 3, and 4, we have*

$$\mathbb{E}[\text{Recall}] := \mathbb{E}[\frac{1}{n_1} \#\{x_i \in X_{real}^1 : x_i \text{ identifiable}\}] \geq L_{id}, \tag{3}$$

*where $L_{id}$ is defined in Equation 2.*

*Sketch Proof.* By Lemma 2, we have $\Pr\{x_i \text{ identifiable}\} \geq L_{id}$ for every $i$, so

$$\mathbb{E}[\text{Recall}] = \frac{1}{n_1} \sum_{i=1}^{n_1} \Pr\{x_i \text{ identifiable}\} \geq L_{id}.$$

$\square$

| Dataset | $r$ | $n$ | $d$ | DistinSMOTE | | ReconSMOTE | |
|---|---|---|---|---|---|---|---|
| | | | | (Precision) | (Recall) | (Precision) | (Recall) |
| higgs | 25 | 47,976 | 28 | $1.00 \pm 0.00$ | $1.00 \pm 0.00$ | $1.00 \pm 0.00$ | $1.00 \pm 0.00$ |
| miniboone | 25 | 96,690 | 50 | $1.00 \pm 0.00$ | $1.00 \pm 0.00$ | $1.00 \pm 0.00$ | $1.00 \pm 0.00$ |

Table 5: Privacy attacks vs. augmented/synthetic data with high-dimensional data.

| Dataset | $r$ | $n$ | $d$ | DistinSMOTE | | ReconSMOTE | |
|---|---|---|---|---|---|---|---|
| | | | (num, cat) | (Precision) | (Recall) | (Precision) | (Recall) |
| cardio | 25 | 7,256 | $(5, 6)$ | $1.00 \pm 0.00$ | $1.00 \pm 0.00$ | $1.00 \pm 0.00$ | $0.98 \pm 0.00$ |
| churn | 25 | 8,269 | $(5, 5)$ | $1.00 \pm 0.00$ | $1.00 \pm 0.00$ | $1.00 \pm 0.00$ | $0.98 \pm 0.01$ |

Table 6: Privacy attacks vs. augmented/synthetic data with mixed-type data.

| Dataset | $r$ | DistinSMOTE | | ReconSMOTE | |
|---|---|---|---|---|---|
| | | (Precision) | (Recall) | (Precision) | (Recall) |
| ecoli | 8.6 | $1.00 \pm 0.00$ | $1.00 \pm 0.00$ | $1.00 \pm 0.00$ | $0.43 \pm 0.06$ |
| abalone | 9.7 | $1.00 \pm 0.00$ | $1.00 \pm 0.00$ | $1.00 \pm 0.00$ | $0.42 \pm 0.02$ |
| car_eval_34 | 12 | $1.00 \pm 0.00$ | $1.00 \pm 0.00$ | $1.00 \pm 0.00$ | $0.61 \pm 0.02$ |
| solar_flare_m0 | 19 | $1.00 \pm 0.00$ | $1.00 \pm 0.00$ | $1.00 \pm 0.00$ | $0.40 \pm 0.02$ |
| car_eval_4 | 26 | $1.00 \pm 0.00$ | $1.00 \pm 0.00$ | $1.00 \pm 0.00$ | $0.60 \pm 0.00$ |
| yeast_me2 | 28 | $1.00 \pm 0.00$ | $1.00 \pm 0.00$ | $1.00 \pm 0.00$ | $0.58 \pm 0.02$ |
| mammography | 42 | $1.00 \pm 0.00$ | $1.00 \pm 0.00$ | $1.00 \pm 0.00$ | $0.61 \pm 0.01$ |
| abalone_19 | 130 | $1.00 \pm 0.00$ | $1.00 \pm 0.00$ | $1.00 \pm 0.00$ | $0.49 \pm 0.01$ |
| average | | $1.00 \pm 0.00$ | $1.00 \pm 0.00$ | $1.00 \pm 0.00$ | $0.52 \pm 0.02$ |

Table 7: Privacy attacks vs. augmented/synthetic data by SMOTE with $k = 2$.

# B    LOWER BOUNDS OF ReconSMOTE RECALL VISUALIZATIONS

To complement the theoretical results in Section 4.3 (approximate bound, $A_{id}$) and Appendix A (exact bound, $L_{id}$), we visualize the bounds under different conditions.

We start by showing the ratio between the exact bound and the approximate bound as a function of the imbalance ratio $r$ in Figure 4. When $\alpha = 0$ (Figure 4a), the approximate bound closely matches the exact one for all $k$, especially for larger imbalance ratio. In contrast, even a small amount of mutuality ($\alpha = 0.1$; Figure 4b) introduces a noticeable deviation of the lower bound.

Next, in Figure 5, we focus on the exact bound $L_{id}$ under varying imbalance ratios $r$ and neighborhood sizes $k$ (with $n_1 = 100$ and $\alpha = 0.5$ fixed). Figure 5a shows that $L_{id}$ steadily increases as the oversampling ratio $r$ rises. For small $k$, even a moderate oversampling ratio results in significant identifiability. However, for larger $k$, a higher oversampling ratio is required. The heatmap in Figure 5b clearly illustrates this interaction. In the upper-left area, where $k$ is small and $r$ is large, $L_{id}$ quickly approaches 1. This indicates almost perfect identifiability. In contrast, in the lower-right area, where $k$ is large and $r$ is small, $L_{id}$ is close to zero. This suggests that the reconstructed edges are not dense enough to reach high identifiability. Overall, these plots confirm the trade-off: identifiability improves with oversampling, but its efficiency depends strongly on the neighborhood parameter $k$.

Finally, Figure 6 presents the exact bound $L_{id}$ as a function of $\alpha$ for several imbalance ratios $r$. The curves illustrate the sensitivity of $L_{id}$ to the graph structure. For example, when $r = 10$, small increases in $\alpha$ would lead to substantial changes in $L_{id}$, highlighting how mutuality in the KNN graph strongly influences privacy leakage.

# C    DistinSMOTE AND ReconSMOTE WITH RELAXED ASSUMPTIONS

In this section, we present results for our attacks under relaxed assumptions – on high-dimensional data (Table 5), mixed-type data (Table 6), SMOTE with $k = 2$ (Table 7), and perturbed data (Table 8 and 9). For the first two experiments, we use datasets different from the eight main datasets in Table 2. Namely, we use higgs and miniboone from OpenML (Vanschoren et al., 2014) as high-dimensional data, and cardio and churn from Kaggle as mixed-type data. These datasets are used in relevant prior work (Kotelnikov et al., 2023). For all datasets, the minority class is undersampled so the imbalance is 25. The results of the first three experiments are discussed in Section 6.

| DistinSMOTE | SMOTE augmented data w/ noise per column (Prec./Rec.) | | | | | | |
|---|---|---|---|---|---|---|---|---|
| | noise $= 10^{-10}$ | | noise $= 10^{-7}$ | | noise $= 10^{-5}$ | | noise $= 10^{-3}$ | |
| tol. $= 10^{-10}$ | **0.96** | 1.00 | 0.88 | 1.00 | 0.04 | 1.00 | 0.04 | 1.00 |
| tol. $= 10^{-7}$ | 0.91 | 0.99 | **0.91** | 0.99 | 0.07 | 1.00 | 0.04 | 1.00 |
| tol. $= 10^{-5}$ | 0.41 | 0.80 | 0.37 | 0.78 | **0.07** | 0.93 | 0.04 | 0.92 |
| tol. $= 10^{-3}$ | 0.09 | 0.31 | 0.09 | 0.31 | 0.06 | 0.82 | **0.06** | 0.80 |

Table 8: `DistinSMOTE` vs. perturbed augmented data, on yeast_me2.

| ReconSMOTE | SMOTE synthetic data w/ noise per column (Prec./Rec.) | | | | | | |
|---|---|---|---|---|---|---|---|---|
| | noise $= 10^{-10}$ | | noise $= 10^{-7}$ | | noise $= 10^{-5}$ | | noise $= 10^{-3}$ | |
| tol. $= 10^{-10}$ | **1.00** | 1.00 | 0.86 | 0.96 | 0.00 | 0.00 | 0.00 | 0.00 |
| tol. $= 10^{-7}$ | 1.00 | 1.00 | **0.96** | 1.00 | 0.00 | 0.00 | 0.00 | 0.00 |
| tol. $= 10^{-5}$ | 0.99 | 0.97 | 0.85 | 0.84 | **0.71** | 0.10 | 0.00 | 0.00 |
| tol. $= 10^{-3}$ | 0.63 | 0.61 | 0.55 | 0.53 | 0.10 | 0.09 | 0.00 | 0.00 |

Table 9: `ReconSMOTE` vs. perturbed synthetic data, on yeast_me2.

Next, we discuss the results of the forth experiment – running our attacks on perturbed SMOTE data.

**SMOTE with Perturbed Linear Interpolation.** We test the robustness of our attacks on perturbed data by adding column-wise noise in the range $\{10^{-10}, 10^{-7}, 10^{-5}, 10^{-3}\}$, ensuring that no synthetic record lies exactly on the line between its generating real records (see Table 8 and 9). Similarly, we use tolerance levels in the same range for detecting lines/intersections within our attacks. We use the yeast_me2 dataset from our main experiments (Table 2).

Perhaps surprisingly, both `DistinSMOTE` and `ReconSMOTE` remain highly effective when the added noise is small ($\leq 10^{-7}$), achieving near-perfect performance. This shows that our attacks can generalize to SMOTE variants that use non-strictly linear interpolation. However, the performance of both attacks drops sharply when larger noise is injected – though such noise levels would likely degrade downstream utility as well. Across all noise settings, we observe a consistent trend: for each noise level, both attacks achieve their best precision (the more important metric, as already discussed) when the tolerance parameter matches the injected noise level.

Finally, we relax another assumption – running our attacks without prior knowledge of $k$ and $r$.

**SMOTE with Unknown $k$ and $r$.** We describe heuristics for running `DistinSMOTE` and `ReconSMOTE` without knowing SMOTE's parameters $k$ and $r$. In practice, precise estimates are unnecessary: the neighborhood search only needs to be wide enough to include the real records that generated a given (synthetic) record. Overestimating $k$ or $r$ does not reduce precision/recall, it only increases runtime. Thus, a simple strategy would be to skip parameter estimation altogether and simply use a large neighborhood search (e.g., 10-25% of the dataset).

To estimate $k$ or $r$ more accurately, the adversary can reuse the same sub-procedures employed in the attacks. For a given record, a large neighborhood search (e.g., 10-25% of the dataset) identifies all neighbors on the same line. For augmented data, the adversary can locate an endpoint (a real record), run a second search around it, and infer: i) the number of lines pointing to this record (an overestimate of $k$), and ii) the number of records per line (a rough approximation of $r/k$). For synthetic data, the adversary can instead detect three intersecting lines around the record, identify their intersection (a real record), and proceed as above. Repeating this procedure and averaging yields stable estimates.

