# OpenReview forum: "SMOTE and Mirrors: Exposing Privacy Leakage from Synthetic Minority Oversampling"
_ICLR.cc/2026/Conference — ICLR 2026 Poster_

### Official Review · Reviewer_Jfyn · 2025-10-27

**Soundness:** 2
**Presentation:** 3
**Contribution:** 3
**Rating:** 4
**Confidence:** 3

**Summary:**

The authors study the privacy leakage of SMOTE in the cases of data augmentation on imbalanced datasets with minority classes, and synthetic data generation. They propose two privacy attacks DistinSMOTE and ReconSMOTE, both achieving perfect precision while naive methods completely fail to detect any privacy leakage. The authors also claim to be the first to run a membership inference attack (MIA) against SMOTE. The state-of-the-art MIAs achieve high AUC, especially in the synthetic data generation case, which provides further evidence that SMOTE is fundamentally non-private.

**Strengths:**

* The paper studies an important problem, namely the privacy leakage of SMOTE.
* The paper proposes two privacy attacks DistinSMOTE (Algorithm 2) and ReconSMOTE (Algorithm 3) for the cases of data augmentation and synthetic data generation, respectively.
* The strength of the proposed privacy attacks are supported by both theoretical results (Theorems 1 - 4) and empirical evidence (Section 4).

**Weaknesses:**

* The proofs can be written more clearly and carefully. There are some gaps in the proofs and they are somewhat informal.
* There are no correctness proofs for algorithms (2 & 3), and the proofs of the theorems (1 - 4) are not based on the steps of algorithms (2 & 3).

**Questions:**

* Wouldn't the proof of Theorem 2 also require Assumption 4 (which is found in appendix A) to support the claim "the only point lying on three or more such lines is the shared real endpoint."?
* It can be stated more clearly in both the statement and proof of Theorem 3, that the segments/edges are directed, which is required for computing the probabilities.
* In Theorems 3 and 4, the proofs rely on the conclusion that each $C_{i j}$ is distributed according to the binomial distribution. This skips a step in the proof that the vector of all $C_{i j}$ has a multinomial distribution with $\sum_{i j} C_{i j} = n_0 - n_1$ which is exactly the number of synthesized points, and then we can conclude that $C_{i j}$ are marginally binomial.
* The proofs can be made stronger and more formal if they were based on the steps of the algorithms.

---

> ### Author Response · Authors · 2025-11-20
> **Authors Response to Reviewer `Jfyn`**
>
> We thank Reviewer `Jfyn` for their thoughtful review and for recognizing/highlighting the strengths of our paper and for rigorously going through our Theorems. In our overall message, we provide a summary of our additional experiments and revisions. Before we address the reviewer's specific concerns, we would like to highlight that:
> * Unlike DCR and previous privacy attacks (Stadler et al., 2022; Houssiau et al., 2022; Annamalai et al., 2024b) on tabular synthetic data, to the best of our knowledge, we are the first to propose a new attack with theoretical analysis.
> * Our empirical experiments back the theoretical analysis, and we achieve perfect precision and nearly perfect recall on all main datasets.
>
> > The proofs can be written more clearly and carefully. There are some gaps in the proofs and they are somewhat informal.
> > The proofs can be made stronger and more formal if they were based on the steps of the algorithms.
>
> Theorem 1-3 were originally meant to establish the correctness and analytical properties of the distinguishing and reconstruction rules in Algorithms 2 and 3, without necessarily following every line of the algorithms. The Algorithms include additional implementation details, such as local search heuristics, which speed up computation but do not change the underlying rules.
>
> To make the proofs more clear, robust, and formal, we updated them in the revised paper. Specifically, in Section 4.2 and 4.3 and Appendix A, we added introductory sentences, intermediate steps, explicitly stated all assumptions, and linked Theorems’ proof steps to the corresponding lines in the Algorithms.
>
> > There are no correctness proofs for algorithms (2 & 3).
>
> Theorem 1-3 establish the correctness and analytical properties of the distinguishing and reconstruction rules in Algorithms 2 and 3. Specifically, they analyze precision and recall, which are the key metrics for evaluating the success of these privacy attacks.
>
> > Wouldn't the proof of Theorem 2 also require Assumption 4 (which is found in appendix A) to support the claim "the only point lying on three or more such lines is the shared real endpoint."?
>
> Assumption 2 (global non-collinearity) is stronger and directly implies Assumption 4 (local non-degeneracy): if no three real points are collinear, then any two lines drawn from a real point to two of its neighbors must have different directions, so the local non-degeneracy holds automatically. Thus, Theorem 2 could be proved under Assumptions 1 and 4 alone. We nevertheless use Assumption 2 for Theorem 2 for consistency and clarity across Theorems 1-3 in the main paper.
>
> In the revised paper, we clarified which assumptions each Theorem uses and added a short explanation of the relationship between Assumption 2 and 4 in Appendix A.
>
> > It can be stated more clearly in both the statement and proof of Theorem 3, that the segments/edges are directed, which is required for computing the probabilities.
>
> In the revised paper, we clarified this by explicitly noting the use of directed edges and a Poisson approximation (which ignores overlapping neighbor relations in the SMOTE graph) just before introducing Theorem 3 in Section 4.3.
>
>  > In Theorems 3 and 4, the proofs rely on the conclusion that each C_ij, is distributed according to the binomial distribution. This skips a step in the proof that the vector of all C_ij has a multinomial distribution with S_ij C_ij = n_0 – n_j which is exactly the number of synthesized points, and then we can conclude that C_ij are marginally binomial.
>
> In the revised paper, we clarified this in both Theorem 3 in Section 4.3 and Theorem 4 in Appendix A.

---

> > ### Comment · Reviewer_Jfyn · 2025-11-27
> >
> > Thank you for incorporating my comments into the revision. I have raised my score.

---

### Official Review · Reviewer_grKo · 2025-10-28

**Soundness:** 3
**Presentation:** 4
**Contribution:** 2
**Rating:** 4
**Confidence:** 4

**Summary:**

The paper investigates the empirical privacy of the widely used data augmentation/synthetic data generation method, SMOTE. They argue that common empirical privacy mettrics such as DCR severely underestimate privacy leakage of SMOTE and introduce two new attacks that exploit SMOTES geometric interpolation to perfectly distinguish real from synthetic samples and reconstruct minority records from synthetic data. These attacks have almost perfect recall on a number of benchmark datasets highlighting the privacy problems of relying on SMOTE in sensitive domains.

**Strengths:**

- The paper is clearly and concisely written, making it easy to follow.
- The proposed attacks have strong empirical results across a range of benchmark datasets and attack settings.
- The proposed attack, is as far as I am aware novel in the sense of exploiting SMOTEs interpolation algorithm with the added benefit that it is relatively “lightweight” as it avoids the use of shadow models.

**Weaknesses:**

- The main contribution -- that SMOTE leaks information and is non-private, is not conceptually new. Most practitioners already recognize that SMOTE, based on linear interpolation, offers no privacy guarantees and a long-line of work in tabular diffusion models (which is cited by this paper) even shows this empirically via DCR.
- The paper stops short of offering mitigation strategies of SMOTE, beyond briefly mentioning differential privacy approaches as future work.
- The framing (“SMOTE is inherently non-private”) overemphasizes a result that, while empirically validated, is expected and offers modest new understanding to the synthetic data community without suggesting defenses.

**Questions:**

1. I don’t believe the main finding, that SMOTE is non-private is particularly novel/insightful. It seems well known that SMOTE is not (empirically) private as shown by a number of DCR results in the line of literature on tabular diffusion models such as [2,3] which already note SMOTEs poor privacy characteristics. Therefore the central claim of the paper is that these DCR results "underestimate" the true privacy leakage which again is not a new insight i.e, it is shown in [4]. Can the authors expand on why their work is novel?
2. The claim that “SMOTE remains one of the most widely applied algorithms for medical synthetic data” seems overstated. Kaabachi et al. (2025) report SMOTE as only a small fraction of methods used, with GAN-based approaches dominating. I find it hard to believe the motivation for this paper i.e., 1) SMOTE is used a lot in practice for privacy-sensitive synthetic data and 2) is claimed as a ‘privacy preserving' method.
3. Wouldn’t adding a small amount of noise to the synthetic point generated from SMOTE hinder your proposed attack since the synthetic points will no longer be collinear? The paper should include more results using naive defenses such as this to show it is robust.
4. The datasets used are small-scale in both the number of rows and columns. Can you discuss the the computational scalability for larger, high-dimensional datasets?
5. How realistic is the assumption that k >= 3 for low-dimensional datasets like Abalone where d=6?

[1] Kaabachi, Bayrem, et al. "A scoping review of privacy and utility metrics in medical synthetic data." *NPJ digital medicine* 8.1 (2025): 60.

[2] Kotelnikov, Akim, et al. "Tabddpm: Modelling tabular data with diffusion models." *International conference on machine learning*. PMLR, 2023.

[3] Zhang, Hengrui, et al. "Mixed-type tabular data synthesis with score-based diffusion in latent space." arXiv preprint arXiv:2310.09656 (2023).

[4] Ganev, Georgi, and Emiliano De Cristofaro. "On the Inadequacy of Similarity-based Privacy Metrics: Reconstruction Attacks against``Truly Anonymous Synthetic Data''." (2023).

---

> ### Author Response · Authors · 2025-11-20
> **Authors Response to Reviewer `grKo` (1/2)**
>
> We thank Reviewer `grKo` for their thoughtful review. In our overall message, we provide a summary of our additional experiments and revisions. We now address the reviewer's specific concerns.
>
> > The main contribution -- that SMOTE leaks information and is non-private, is not conceptually new.
> > I don’t believe the main finding, that SMOTE is non-private is particularly novel/insightful.
>
> Respectfully, please note that, while SMOTE was not designed for privacy (as noted in Section 1), it is in fact widely deployed on sensitive data (e.g., in healthcare/finance pipelines and MLaaS systems, as substantiated by the corresponding references in Section 1). However, its privacy properties have not been systematically examined. To the best of our knowledge, no prior research has shown any findings comparable to ours: our attacks do not merely leak information -- they completely break SMOTE, achieving perfect distinguishing and exact reconstruction under minimal assumptions. Thus, the contribution is not the observation that "SMOTE leaks information," but the first rigorous, theoretically grounded evidence (and analysis thereof) that it provides no privacy at all in practice.
>
>
> > Most practitioners already recognize that SMOTE, based on linear interpolation, offers no privacy guarantees and a long-line of work in tabular diffusion models (which is cited by this paper) even shows this empirically via DCR.
>
> > It seems well known that SMOTE is not (empirically) private as shown by a number of DCR results in the line of literature on tabular diffusion models such as [2,3] which already note SMOTEs poor privacy characteristics. Therefore the central claim of the paper is that these DCR results "underestimate" the true privacy leakage which again is not a new insight i.e, it is shown in [4].
>
> Respectfully, please note that the diffusion-model papers (Kotelnikov et al., 2023; Zhang et al., 2024; Pang et al., 2024; Mueller et al., 2025) do NOT demonstrate that SMOTE has meaningful privacy leakage. In this line of work, SMOTE is used only as a baseline within the following "SMOTE+DCR" workflow:
> 1. Propose a new tabular diffusion model (not designed for privacy)
> 2. Use DCR as the sole privacy metric, observe DCR(diffusion model) > DCR(SMOTE)
> 3. Conclude that the diffusion model is "more private" than SMOTE
>
> Moreover, in prior research:
> * Critiques of DCR (Ganev & De Cristofaro, 2025; Yao et al., 2025) show that DCR is not a reliable privacy metric and does not correlate with MIAs (the gold-standard measure of privacy).
> * Sidorenko & Tiwald (2025) even show that DCR can rank diffusion models (TABDDPM (Kotelnikov et al., 2023) and CDTD (Mueller et al., 2025)) as LESS private than SMOTE.
>
> Nevertheless, none of these (or other) works analyzes SMOTE itself or quantifies its leakage. Our paper directly addresses this gap and raises questions about whether the "SMOTE + DCR" workflow is fundamentally flawed. More precisely/to this end, our work:
> * completely break SMOTE (perfect distinguishing and exact reconstruction),
> * comes with theoretical guarantees, and
> * reveals orders of magnitude more leakage than DCR reports.
>
> To summarize, our contribution goes far beyond the statements that "SMOTE leaks information" and "DCR underestimates leakage." We provide the first concrete evidence that the ML community’s standard privacy-assessment workflow ("SMOTE+DCR") is misleading: SMOTE is dramatically less private than previously recognized, and DCR fails to detect this. This directly challenges step 2) of the SMOTE+DCR workflow discussed above (if a model appears slightly more or less "private" than a clearly non-private baseline under a flawed privacy metric, what meaningful conclusion can be drawn?).
>
> We clarified and emphasized this "SMOTE + DCR" motivation in Sections 1, 3, and 7 of the revised paper.
>
> > Can the authors expand on why their work is novel?
>
> To summarize, our work makes several first-time contributions:
> * We show that widely used privacy metrics like DCR barely detect any leakage from SMOTE.
> * We run state-of-the-art tabular MIAs against SMOTE and demonstrate they are highly effective.
> * We introduce novel, minimal-assumption, and efficient attacks on SMOTE, achieving perfect distinguishing and exact reconstruction.
>
> Importantly, unlike DCR and prior privacy attacks on tabular synthetic data, our attacks are accompanied by rigorous theoretical analysis. These findings have several key implications for the ML (and medical/finance) community:
> 1. SMOTE is fundamentally non-private.
> 2. Minority records are disproportionately at risk.
> 3. SMOTE, combined with naive privacy metrics like DCR, cannot serve as a reliable baseline for assessing the privacy of other generative models.
> 4. Performance gains from oversampling must be weighed vs. privacy risks.

---

> > ### Author Response · Authors · 2025-11-20
> > **Authors Response to Reviewer `grKo` (2/2)**
> >
> > > I find it hard to believe the motivation for this paper i.e., 1) SMOTE is used a lot in practice for privacy-sensitive synthetic data and 2) is claimed as a ‘privacy preserving' method.
> >
> > Our intention has never been to argue that SMOTE itself is a privacy-preserving method. Instead, we explain that while SMOTE was not designed for privacy (as noted in Section 1), 1) it is in fact widely deployed on sensitive data (e.g., healthcare/finance pipelines and MLaaS systems, as substantiated by our references in Section 1) as a pre-processing/augmentation tool, and 2) it is commonly used as a baseline when evaluating the privacy of other generative models for synthetic data, such as diffusion models. We clarified this motivation in Section 1 of the revised paper.
> >
> > > The claim that “SMOTE remains one of the most widely applied algorithms for medical synthetic data” seems overstated. Kaabachi et al. (2025) report SMOTE as only a small fraction of methods used, with GAN-based approaches dominating.
> >
> > We updated the statement in Section 1 of the revised paper to clarify that "SMOTE is also applied for medical synthetic data." Additionally, while we already cite 10+ peer-reviewed papers using SMOTE in medicine for pre-processing/augmentation, many more relevant citations could be included.
> >
> > > The paper stops short of offering mitigation strategies of SMOTE, beyond briefly mentioning differential privacy approaches as future work.
> > > Wouldn’t adding a small amount of noise to the synthetic point generated from SMOTE hinder your proposed attack since the synthetic points will no longer be collinear? The paper should include more results using naive defenses such as this to show it is robust.
> > > The framing (“SMOTE is inherently non-private”) overemphasizes a result that, while empirically validated, is expected and offers modest new understanding to the synthetic data community without suggesting defenses.
> >
> > We evaluate our attacks on the standard SMOTE implementation from imblearn, which is widely used and has nearly 1 million downloads per month. We discussed potential defenses in Section 7 of the revised paper:
> > * Non-linear variants: SMOTE variants that do not rely on linear interpolation (e.g., G-SMOTE, GI-SMOTE) would prevent our attacks.
> > * Differential privacy: using a DP version of SMOTE would also mitigate our attacks.
> >
> > However, SMOTE and its interpolation-based variants are intrinsically non-private and are broadly vulnerable to MIAs, which achieve AUC = 0.93 on 100 target records from synthetic datasets. Even if naive ad-hoc defenses, such as adding a small amount of noise, disrupt our line-based attacks, MIAs would still succeed, demonstrating that SMOTE’s privacy risk is fundamental.
> >
> > Therefore, SMOTE should not be used for privacy-preserving synthetic data generation or as a privacy baseline. We clarified this position in Section 7 of the revised paper.
> >
> > > The datasets used are small-scale in both the number of rows and columns. Can you discuss the the computational scalability for larger, high-dimensional datasets?
> >
> > The main datasets from our experiments are standard benchmarks for evaluating models on imbalanced data like SMOTE and are used in a recent ICML paper (Rosenblatt et al., 2025).
> >
> > We provide the computation complexity of both attacks, showing that they would scale well both with the number of rows/columns. Additionally, our theoretical analysis of ReconSMOTE recall shows that it is independent of n and d. To empirically verify these two hypotheses, we ran our attacks on two large, high-dimensional datasets (up to 96,690 records and 50 features) in Section 6 of the revised paper, achieving perfect results (see Table 5 in the revised paper). Both attacks complete in under 8 minutes per dataset, which is highly practical.
> >
> > Additionally, we evaluated our attacks on two mixed-type datasets in Section 6 of the revised paper. We obtain perfect precision and very high recall (see Table 6 in the revised paper).
> >
> > > How realistic is the assumption that k >= 3 for low-dimensional datasets like Abalone where d=6?
> >
> > We note that k and d are independent: k is the number of neighbors SMOTE uses to generate synthetic points, while d  is the dimensionality of the dataset. The default value of k in SMOTE is 5, and it makes more sense to change with n, rather than d. Using k >= 3 is perfectly reasonable even for low-dimensional datasets such as Abalone (d=6) or simple 2d toy datasets.

---

> > > ### Comment · Reviewer_grKo · 2025-11-27
> > >
> > > I would like to thank the authors for their rebuttal and the revised version of their work.
> > >
> > > > We updated the statement in Section 1 of the revised paper to clarify that "SMOTE is also applied for medical synthetic data." Additionally, while we already cite 10+ peer-reviewed papers using SMOTE in medicine for pre-processing/augmentation, many more relevant citations could be included
> > >
> > > As you noted, the majority of these cited medical applications use SMOTE only as a pre-processing step before training an ML model. In almost all of these cases, privacy is not a focus or a concern. Moreover, these applications do not involve data release, meaning the privacy threat model differs significantly from the one studied in this paper. In such settings, any leakage is in the downstream ML model trained on SMOTE-augmented data rather than through released synthetic points. This is why I remain unconvinced by the original motivation presented in the paper.
> > >
> > > > Respectfully, please note that the diffusion-model papers (Kotelnikov et al., 2023; Zhang et al., 2024; Pang et al., 2024; Mueller et al., 2025) do NOT demonstrate that SMOTE has meaningful privacy leakage.
> > > > We provide the first concrete evidence that the ML community’s standard privacy-assessment workflow ("SMOTE+DCR") is misleading: SMOTE is dramatically less private than previously recognized, and DCR fails to detect this.
> > >
> > > The framing above and the reworded motivation in the revised paper that focuses more on SMOTE being a poor privacy bar for the tabular difusion literature/ML community is quite convincing, moreso than your suggested applications in medicine/finance.
> > >
> > > > We evaluate our attacks on the standard SMOTE implementation from imblearn, which is widely used and has nearly 1 million downloads per month. We discussed potential defenses in Section 7 of the revised paper:
> > > > Non-linear variants: SMOTE variants that do not rely on linear interpolation (e.g., G-SMOTE, GI-SMOTE) would prevent our attacks.
> > > > Differential privacy: using a DP version of SMOTE would also mitigate our attacks.
> > >
> > > Could you clarify whether you believe the attack would fully fail under non-linear SMOTE variants, or whether partial matches may occur? The paper would be substantially stronger if it included comparisons against simple baseline defenses, such as (1) non-linear SMOTE variants and (2) naive noise addition.
> > >
> > > > Additionally, we evaluated our attacks on two mixed-type datasets in Section 6 of the revised paper. We obtain perfect precision and very high recall (see Table 6 in the revised paper).
> > >
> > > I appreciate that addition of Section 6 which directly addresses my concerns (alongside those of others reviewers), demonstrating stronger robustness.

---

> ### Author Response · Authors · 2025-12-03
> **Authors Response to Reviewer `grKo`**
>
> We thank Reviewer `grKo` for their continued engagement, acknowledging our reply/revisions, and their further helpful suggestions. We are also glad the reviewer finds our revised motivation and additional experiments convincing, and that they address the most important concerns. Next, we address the remaining two concerns: 1) SMOTE used as a preprocessing tool, and 2) SMOTE with naive noise addition.
>
> > The majority of these cited medical applications use SMOTE only as a pre-processing step before training an ML model. In almost all of these cases, privacy is not a focus or a concern. …  In such settings, any leakage is in the downstream ML model trained on SMOTE-augmented data rather than through released synthetic points.
>
> We completely agree, in these applications (medical/finance), SMOTE is used as a preprocessing step, and it seems privacy is not explicitly considered. However, because these domains involve highly sensitive data, we believe privacy is still a concern. We also agree that any leakage in these settings would come from the downstream ML model trained on SMOTE-augmented data. This is exactly why Section 5.1 evaluates MIA: we show that models trained on SMOTE-augmented data become more vulnerable (by 17% on average, Table 3) compared to the real data only.
>
> Additionally, in realistic deployments, internal analysts/data scientists/developers are likely to have access to the augmented data (since privacy is not treated as a threat). In such cases, DistinSMOTE could be run directly and would perfectly distinguish real/synthetic records.
>
> > Could you clarify whether you believe the attack would fully fail under non-linear SMOTE variants, or whether partial matches may occur? The paper would be substantially stronger if it included comparisons against simple baseline defenses, such as (1) non-linear SMOTE variants and (2) naive noise addition.
>
> In the newly revised paper, we ran additional experiments, evaluating our attacks on SMOTE data with added noise so the generated synthetic records are not exactly on the line between their generating real records (see Table 8 and 9 in Appendix C in the newly revised paper). We show that both DistinSMOTE and ReconSMOTE are robust (perhaps surprisingly) when the added noise per column is relatively low (<=1e-7). DistinSMOTE has partial matches even for larger added noise. This leads us to believe that our attacks could have some partial matches on non-linear SMOTE variants, but likely very low, even though we leave generalizing our attacks to other non-linear SMOTE variants to future work (we are confident our attacks would be very successful on all SMOTE variants in imblearn).

---

### Official Review · Reviewer_cZmi · 2025-10-28

**Soundness:** 2
**Presentation:** 4
**Contribution:** 4
**Rating:** 4
**Confidence:** 4

**Summary:**

The paper studies the privacy vulnerability of SMOTE, a technique for augmenting imbalanced data or generating synthetic data. The paper presents two attacks against SMOTE that exploit the fact that SMOTE generates new points by interpolating real points. The first can perfectly distinguish SMOTE-generated datapoints from real datapoints in an augmented dataset under suitable assumptions. The second reconstructs real points from SMOTE-generated data. Both attacks are empirically evaluated and compared with existing MIAs and distance-based attacks. Both attacks have extremely high accuracy, while the baselines do not.

**Strengths:**

The paper clearly shows that SMOTE is not suitable to any privacy-sensitive application, whereas previous evaluations, especially distance-based ones, are less convincing. The writing of the paper is good, and the principles of the attacks are explained clearly.

While SMOTE is not considered a serious synthetic data generation method in the machine learning community, the citations in the paper show that SMOTE is seriously considered in other important research communities. The results of the paper show that it should not be. The new attacks are so accurate that releasing SMOTE-generated synthetic data is not far from releasing real data, and if releasing real data is acceptable, generating synthetic data does not make sense.

The results also demonstrate a flaw in most empirical privacy evaluations. Typical evaluations only use generic metrics like DCR or attacks like MIAs that can be applied to many methods without modifications. However, these do not rule out the possibility that an attack targeted at a specific method could lead to a much higher privacy loss than suggested by generic metrics, which is what this paper shows for SMOTE.

In summary, though the scope of the paper is limited to SMOTE, there are enough wider implications in the work to warrant publication in ICLR once some technical issues with the theory (described below) are resolved.

**Weaknesses:**

There are some incorrect technicalities in Theorems 1 and 2. For Theorem 1, it is possible that a SMOTE-generated point happens to be collinear with two real points other than the points it was interpolated from. In this case, one of the unrelated real points could be the midpoint of the line between the three, and it would be classified as a synthetic point, while the generated points would be classified as real.

Theorem 2 has a similar issue: it is possible that three generated points from different real points just happen to be collinear. If that happens multiple times, ReconSMOTE could reconstruct points that are not real. Also in Theorem 2, global collinearity does not guarantee that any intersection of more than three lines is a real point. It is possible to draw three lines intersecting at one point, and pick 2 points from each line to be real points, such that no three of the 6 real points are collinear.

However, all of these issues are very unlikely to happen in practice, so they can likely be fixed by adding some reasonable assumption. For example, I would not be surprised if the probability of these happening is 0 when the real datapoints are sampled from a continuous distribution in $d$-dimensions.

In addition, the proofs of Theorems 1 and 2 should show that looking at a subset of nearest neighbours of a point is sufficient instead of looking at all points (line 9 in Algorithm 1 and line 8 in Algorithm 2).

**Questions:**

- Lines 181-182: how can the number of neighbours and imbalance ratio be approximated?
- Would setting $k = 2$ be an effective defence against the attacks?
- Do the attacks work if applied against one-hot encoded categorical data? Does SMOTE even make sense in this case?

---

> ### Author Response · Authors · 2025-11-20
> **Authors Response to Reviewer `cZmi`**
>
> We thank Reviewer `cZmi` for their thoughtful review and for recognizing/highlighting the strengths and wide applicability of our paper. In our overall message, we provide a summary of our additional experiments and revisions. We now address the reviewer's specific concerns.
>
> > For Theorem 1, it is possible that a SMOTE-generated point happens to be collinear with two real points other than the points it was interpolated from. In this case, one of the unrelated real points could be the midpoint of the line between the three, and it would be classified as a synthetic point, while the generated points would be classified as real.
>
> We acknowledge that this scenario is theoretically possible and would indeed reduce the attacks’ performance. However, we work in high-dimensional continuous datasets with high numerical precision (10e-12), so the probability that a SMOTE-generated point lies exactly collinear with two unrelated real points is effectively zero. Importantly, this situation never occurs in any of our empirical experiments (precision is consistently perfect), indicating that it is not a practical concern. We explained this in Section 7 of the revised paper.
>
> > Theorem 2 has a similar issue: it is possible that three generated points from different real points just happen to be collinear. If that happens multiple times, ReconSMOTE could reconstruct points that are not real. Also, in Theorem 2, global collinearity does not guarantee that any intersection of more than three lines is a real point. It is possible to draw three lines intersecting at one point, and pick 2 points from each line to be real points, such that no three of the 6 real points are collinear.
>
> We agree that this scenario is theoretically possible but even less likely, as it would need to occur repeatedly and in a coordinated manner across multiple lines. As with the previous case, this situation never occurs in any of our empirical experiments, and our reconstruction precision remains perfect. Thus, while these cases are theoretically valid edge scenarios (though with a statistical probability of 0), they do not occur in practice and do not undermine the empirical relevance of our attacks.
>
> > In addition, the proofs of Theorems 1 and 2 should show that looking at a subset of nearest neighbours of a point is sufficient instead of looking at all points (line 9 in Algorithm 1 and line 8 in Algorithm 2).
>
> Theorem 1-3 establish the correctness and analytical properties of the distinguishing and reconstruction rules in Algorithms 2 and 3, without necessarily following every line of the algorithms. The Algorithms include additional implementation details, such as local search heuristics, which speed up computation but do not change the underlying rules.
>
> To make the proofs more clear, robust, and formal, we updated them in the revised paper. Specifically, in Section 4.2 and 4.3 and Appendix A, we linked Theorems’ proof steps to the corresponding lines in the Algorithms.
>
> > Lines 181-182: how can the number of neighbours and imbalance ratio be approximated?
>
> We provided a heuristic in Appendix C of the revised paper showing how the adversary can estimate k and r directly from the augmented/synthetic data. In short, a wide neighborhood search (e.g., 10-25% of the dataset) recovers a real/intersection point, its associated line structures, and the counts of synthetic records along these lines, yielding a stable upper bound on k and a rough estimate of r/k.
>
> Alternatively, the adversary may skip parameter estimation entirely and simply use a large neighborhood search (e.g., 10-25% of the dataset).
>
> > Would setting k=2 be an effective defence against the attacks?
>
> We ran an additional experiment with k=2 on all main datasets in Section 6 of the revised paper (i.e., relaxing Assumption 3). DistinAttack remains fully effective, achieving perfect precision and recall (see Table 7 in the revised paper). ReconAttack also maintains perfect precision, while its recall drops by 39% to ~0.52 -- an expected consequence of each real record participating in fewer lines, making it harder to reach the support threshold. Nevertheless, this still constitutes a substantial privacy breach, so k=2 should not be considered an effective defense.
>
> > Do the attacks work if applied against one-hot encoded categorical data? Does SMOTE even make sense in this case?
>
> We ran an additional experiment on two mixed-type datasets in Section 6 of the revised paper (i.e., relaxing Assumption 1). We use SMOTE-NC, which operates on mixed data, to generate synthetic data, and, for the attacks, we simply ignore categorical features and operate on the continuous ones. We still obtain perfect precision and very high recall on both datasets (see Table 6 in the revised paper). This approach can also be applied to one-hot encoded data.

---

> > ### Comment · Reviewer_cZmi · 2025-11-21
> >
> > Thank you for the response.
> >
> > I agree that the pathological scenarios I highlighted are very unlikely in practice and do not take away from the significance of your results. However, as currently stated, Theorems 1-3 are incorrect. If you present a result as a theorem, you need to prove it fully, including all edge cases that the assumptions of the theorem allow, no matter how unlikely they are. If an unlikely edge case makes the theorem incorrect, you should add an explicit assumption that excludes that case, or adjust the conclusion of the theorem.
> >
> > In this case, it probably suffices to add something like "almost surely" or "with probability 1" to the conclusions of the theorems. You may not even need any additional assumptions, since there is already randomness in what points SMOTE chooses, and it seems like the probability of SMOTE choosing a point causing a pathological scenario is zero. But you should verify this argument before adding it to the proofs.
> >
> > This also highlights a weakness of the proofs that Reviewer Jfyn brought up: the proofs are very high-level, even in the revised version, especially for Theorems 1 and 2. Adding more detailed arguments would help showing that there are no more of these pathological edge cases.
> >
> > If writing such detailed proofs would take too much time for the rebuttal, I would also be satisfied if you instead present the theory as heuristic arguments instead of rigorous proofs. The empirical results are strong enough to support the paper's message even without fully rigorous theory.
> >
> > Your response addressed all of my other points very well.

---

> ### Author Response · Authors · 2025-11-22
> **Authors Response to Reviewer `cZmi`**
>
> We thank the reviewer for their continued engagement, the quick reply, and the insightful/helpful suggestions. Below, we address the remaining concerns regarding Theorems 1-3.
>
> > In this case, it probably suffices to add something like "almost surely" or "with probability 1" to the conclusions of the theorems. You may not even need any additional assumptions, since there is already randomness in what points SMOTE chooses, and it seems like the probability of SMOTE choosing a point causing a pathological scenario is zero. But you should verify this argument before adding it to the proofs.
>
> In the newly revised paper, we updated the conclusions of the proofs of Theorem 1 and 2 to explicitly state that they hold with probability 1. Specifically, we added “with probability 1, except for negligible numerical precision effects that do not occur in practice” to the conclusion of the proof of Theorem 1 and “with probability 1” to Theorem 2.
>
> We believe this clarification is appropriate because the probability of pathological scenarios (e.g., a synthetic record lying exactly collinear with non-generating real points, multiple identical synthetic points produced independently from distinct real records, etc.) is zero under Assumption 1 and 2, and the random selection mechanism in SMOTE. Theoretically and statistically speaking, the SMOTE interpolation parameter u is drawn from a continuous uniform distribution, so the event that it generates a degenerate point (i.e., satisfying one of the pathological scenarios) has probability zero. This argument becomes stronger in higher dimensions. Practically speaking, numerical operations are finite, but we use very high numerical precision (10e-12), so the same “with probability 1” guarantee holds empirically as well, consistent with all our experimental results.
>
> Thus, we believe the newly revised theorems now rigorously address all admissible edge cases, while remaining both mathematically precise and practically meaningful.
>
> > the proofs are very high-level, even in the revised version, especially for Theorems 1 and 2. Adding more detailed arguments would help showing that there are no more of these pathological edge cases.
> If writing such detailed proofs would take too much time for the rebuttal, I would also be satisfied if you instead present the theory as heuristic arguments instead of rigorous proofs.
>
> In the newly revised paper, we changed “Proof” to “Sketch Proof” for Theorems 1-4 to better reflect the intended level of rigor. Our aim is to provide a principled theoretical explanation of why the two proposed attacks work, while the paper's main contribution remains empirical: demonstrating significant privacy leakage by SMOTE in practice. (Incidentally, to the best of our knowledge, this is the first work to present a privacy attack on tabular synthetic data with any theoretical analysis.)
>
> Arguably, the sketch proofs justify the labeling and reconstruction rules, state the necessary assumptions, and link to the Algorithms that implement them. We agree that a full line-by-line formalization would be a nice addition and, while possible in principle, that would require space and mathematical machinery we believe to be beyond the scope of this paper and similar work on privacy attacks. We hope that presenting them as sketch proofs strikes the appropriate balance, remaining technically correct while supported by our empirical results.

---

> > ### Comment · Reviewer_cZmi · 2025-11-23
> >
> > Thank you for the update. I think the "with probability 1" caveats should also be in the theorem statements, and that using the word "theorem" is too strong since you only have proof sketches. But the new changes clearly communicate these caveats, so the previous comment is mostly stylistic. As a result, I've increased my score.

---

> ### Author Response · Authors · 2025-11-25
> **Authors Response to Reviewer `cZmi`**
>
> We thank the reviewer once again for their time, engagement, helpful feedback/suggestions, and for substantially increasing their score.
>
> Following the suggestion, we have made another (minor) revision to include the “with probability 1” caveat explicitly in Theorems 1 and 2. Regarding the terminology, we would like to continue referring to these as “Theorems,” as they state the necessary assumptions and formal statements of the labeling/reconstruction rules that our Algorithms implement. Although the proofs are presented as sketches, the theoretical statements are fully specified and supported by our empirical results. We hope the reviewer finds this approach reasonable.

---

### Official Review · Reviewer_G3wE · 2025-11-03

**Soundness:** 3
**Presentation:** 3
**Contribution:** 3
**Rating:** 6
**Confidence:** 3

**Summary:**

This paper presents a systematic privacy analysis of SMOTE (Synthetic Minority Oversampling Technique), one of the most widely used methods for addressing class imbalance in machine learning. The authors show that existing privacy evaluation metrics, including naive classifiers and DCR, completely fail to detect leakage. While standard membership inference attacks achieve a high success rate against SMOTE-generated data, they still substantially underestimate the true extent of privacy leakage. They further introduce two novel geometry-based attacks: DistinSMOTE, which perfectly distinguishes real minority records from synthetic ones in augmented datasets, and ReconSMOTE, which reconstructs the original minority records from only the synthetic data with theoretically perfect precision and high recall. Supported by formal proofs, theoretical bounds, and experiments on eight benchmark datasets, the paper concludes that SMOTE is inherently non-private, exposing the very minority individuals it aims to protect, and calls for a reevaluation of its use in privacy-sensitive applications.

**Strengths:**

- The paper tackles an important and long-overlooked issue in synthetic data privacy by revealing that SMOTE’s interpolation mechanism can inherently leak information about real minority records. This problem is well-motivated, timely, and relevant to both theoretical and applied ML communities.
- The paper is clearly written and visually well-presented. The pseudocode, figures, and tables effectively communicate both the attack logic and empirical findings, making the complex geometric reasoning easy to follow.
- The proposed attacks are original and analytically grounded rather than heuristic. DistinSMOTE and ReconSMOTE exploit the geometric structure of SMOTE in a principled way, offering both theoretical depth and conceptual elegance.
- The experimental results, though limited in data complexity, convincingly demonstrate that existing privacy metrics underestimate leakage while the proposed methods reveal substantial vulnerabilities. This provides strong empirical support for the paper’s central claim.

**Weaknesses:**

- The three core assumptions of this work are too idealized to hold in most real-world settings. Most real datasets (especially in medicine or finance) contain categorical or mixed-type features, violating Assumption 1. Even in continuous data, collinearity or near-collinearity often occurs due to correlated features or duplicated records, which could break Assumption 2.

- Theorem 3’s recall bound relies on overly idealized assumptions of independence and uniform sampling that do not hold in real SMOTE graphs, where neighbor relationships overlap and edges share endpoints. The uniform segment probability ignores heterogeneous data densities, and in high-dimensional spaces, line intersections are rarely exact and must rely on numerical thresholds that introduce errors. As a result, the bound offers useful intuition but overestimates recall under realistic, dependent, and noisy conditions.

- No experiments show how the attacks behave when collinearity, floating-point noise, or categorical features are introduced.

- The datasets used in the experiments are relatively simple and do not fully reflect the complexity or real-world settings of the SMOTE applications discussed in the introduction.

**Questions:**

- Given that many real-world datasets are mixed-type, do you believe these attacks constitute a realistic privacy threat, or mainly a theoretical warning?
- The paragraph describing the assumptions of adversary model is inconsistent (Section 4.1). It first states that the adversary can approximate (k) and (r) from the released dataset, but then claims that no access to public data or published aggregate statistics is required.
- The table reference in line 97 is incorrect; it should refer to Table 2, not Table 1.
- Please also address my concerns raised in the weaknesses section, as resolving them could lead me to raise my score.

---

> ### Author Response · Authors · 2025-11-20
> **Authors Response to Reviewer `G3wE`**
>
> We thank Reviewer `G3wE` for their thoughtful review and for recognizing/highlighting the strengths of our paper. In our overall message, we provide a summary of our additional experiments and revisions. We now address the reviewer's specific concerns.
>
> > The three core assumptions of this work are too idealized to hold in most real-world settings.
> > The datasets used in the experiments are relatively simple and do not fully reflect the complexity or real-world settings of the SMOTE applications discussed in the introduction.
>
> Respectfully, we believe that the three core assumptions align naturally with high-dimensional (continuous) data, are non-restrictive in practice, and, crucially, are satisfied by all datasets in our main experiments. In fact, these datasets are standard benchmarks for evaluating models on imbalanced data like SMOTE and are used in a recent ICML paper (Rosenblatt et al., 2025). Furthermore, we demonstrate that our attacks remain effective even when some assumptions are relaxed (please see the next point).
>
>
> > No experiments show how the attacks behave when collinearity, floating-point noise, or categorical features are introduced.
> > Given that many real-world datasets are mixed-type, do you believe these attacks constitute a realistic privacy threat, or mainly a theoretical warning?
>
> We included an additional experiment on two mixed-type datasets in Section 6 of the revised paper (i.e., relaxing Assumption 1). We use SMOTE-NC, which operates on mixed data, to generate synthetic data, and, for the attacks, we simply ignore categorical features and operate on the continuous ones. We still obtain perfect precision and very high recall on both datasets (see Table 6 in the revised paper).
>
> We also included an additional experiment on large, high-dimensional datasets (up to 96,690 records and 50 features), achieving perfect results (see Table 5 in the revised paper).
>
> Since all of our experiments rely on the default SMOTE/SMOTE-NC implementations in imblearn, and collinearity/floating-point noise is not an issue in any dataset, we believe our attacks represent a realistic and practically meaningful privacy threat, not simply a theoretical warning.
>
> > Theorem 3’s recall bound relies on overly idealized assumptions of independence and uniform sampling that do not hold in real SMOTE graphs, where neighbor relationships overlap and edges share endpoints. … As a result, the bound offers useful intuition but overestimates recall under realistic, dependent, and noisy conditions.
>
> We intentionally use simplified/idealized assumptions in Theorem 3 to provide a compact and interpretable lower bound on recall. These assumptions make the bound looser (i.e., worse) than reality, not tighter. In fact, Theorem 3 underestimates recall because we assume directed/non-overlapping edges, which cause the expected counts on each edge to be underestimated (we need to encounter them twice). Theorem 4 explicitly accounts for mutual (two-way) neighbor relationships in real SMOTE graphs, which increases the effective counts C_ij and yields a tighter, more accurate bound. We clarified these distinctions in Section 4.3 of the revised paper.
>
> Crucially, in Figure 4, both the approximate (green) and exact (orange) bounds are strictly lower and consistently underestimate the empirically observed recall (blue), confirming that neither bound overstates the attack’s effectiveness.
>
> > The paragraph describing the assumptions of adversary model is inconsistent (Section 4.1). It first states that the adversary can approximate (k) and (r) from the released dataset, but then claims that no access to public data or published aggregate statistics is required.
>
> We provided a heuristic in Appendix C of the revised paper showing how the adversary can estimate k and r directly from the augmented/synthetic data. In short, a wide neighborhood search (e.g., 10-25% of the dataset) recovers a real/intersection point, its associated line structures, and the counts of synthetic records along these lines, yielding a stable upper bound on k and a rough estimate of r/k.
>
> Alternatively, the adversary may skip parameter estimation entirely and simply use a large neighborhood search (e.g., 10-25% of the dataset).
>
> > The table reference in line 97 is incorrect; it should refer to Table 2, not Table 1.
>
> The reference to Table 1 is intentional, as it provides a summary/aggregation of our main results, which we then discuss in the subsequent text (lines 101-107 of the revised paper). To avoid confusion, we clarify this in line 99 of the revised paper.

---

> > ### Comment · Reviewer_G3wE · 2025-11-27
> >
> > Thanks for the response and clarification. I will keep my score.

---

### Author Response · Authors · 2025-11-20
**Summary of Additional Experiments and Revisions**

We thank the reviewers for their time, thoughtful reviews, and constructive suggestions. Based on their feedback, we ran four additional experiments and made several changes/clarifications in the revised paper (marked in red/orange):
* In Section 6 (and Appendix C), we run our attacks on two new large, high-dimensional datasets (up to 96,690 records and 50 features), addressing concerns about scalability. (`grKo`)
* In Section 6 (and Appendix C), we run our attacks on two new mixed-type datasets, addressing concerns about operating only on numerical data. (`G3wE`, `cZmi`)
* In Appendix C, we run our attacks against SMOTE with added noise, since reviewers asked whether such naive defenses would negate our attacks. (`grKo`)
* In Appendix C, we run our attacks against SMOTE with k=2, since reviewers asked whether k=2 could serve as a viable defense. (`cZmi`)

Changes/clarifications:
* In Section 1, we clarified the motivation of the paper: 1) SMOTE is used on sensitive data (healthcare/finance) as a pre-processing/augmentation tool, and 2) SMOTE is used as a baseline when evaluating privacy of diffusion models (not as a privacy-preserving technique itself). (`grKo`)
* In Sections 1 (motivation/implications), 3 (related work), and 7 (conclusion), we clarified why the "SMOTE + DCR" workflow is unreliable for assessing the privacy of other generative models. (`grKo`)
* In Section 4, we explained that our attacks’ assumptions are non-restrictive for high-dimensional continuous data and hold in practice. (`G3wE`, `cZmi`)
* In Sections 4.2 and 4.3, we added explanations in the proofs to more closely connect the theorems with the steps of Algorithm 2 and 3. (`cZmi`, `Jfyn`)
* In Sections 4.2 and 4.3, we elaborated that the proofs of Theorem 1 and 2 hold with probability 1, except for negligible numerical precision effects. (`cZmi`)
* In Sections 4.2, 4.3, and Appendix A, we changed the notation of "Proof" to "Sketch Proof" for Theorems 1-4 to better reflect the intended level of rigor. (`cZmi`, `Jfyn`)
* In Section 4.3, we elaborated that Theorem 3 uses directed segments/edges. (`G3wE`, `Jfyn`)
* In Section 4.3, we clarified that the vector of all C_ij follows a multinomial distribution. (`Jfyn`)
* In Section 6, we added explanations of the new experiments on high-dimensional datasets, mixed-type datasets, and SMOTE with k=2. (`G3wE`, `cZmi`, `grKo`)
* In Section 7, we clarified that while numerical instabilities/edge cases are theoretically possible, their probability is effectively zero in practice. (`cZmi`)
* In Section 7, we explained that while region-based SMOTE variants would protect vs. our attacks, they are unlikely to protect vs. MIAs. (`grKo`)
* In Appendix C, we introduced a heuristic for running our attacks without assuming knowledge of k and r. (`G3wE`, `cZmi`)

---

### Author Response · Authors · 2025-12-03
**Summary of Discussion Period**

We wish to thank the reviewers for engaging in the discussion before it was closed due to the security API incident. We are not sure exactly what the most appropriate course of action is after the PC chairs decided to revert the scores, but we hope to clarify the paper's status for the new area chair.

- After the initial reviews, we posted a “global” response, reviewer-specific replies, and made substantial revisions to the paper.
- In short, we emphasized that SMOTE is widely used in real-world, privacy-sensitive settings, not just for oversampling, and clarified that our goal is to show that SMOTE itself can be dangerously non-private. In response to some of the concerns related to assumptions and realism, we clarified that the “pathological” configurations that might break our proofs have essentially zero probability in practice, clarified the role of the Assumptions, and explained how an attacker can infer SMOTE parameters like k and the oversampling ratio directly from the synthetic/augmented dataset. We also added explanations to Theorem 1-4 and their proofs, making direct connections between the proofs and the steps in Algorithms 2-3. Additionally, we ran new/clarified existing experiments, including on larger datasets, mixed-type datasets (via SMOTE-NC), and perturbed datasets. We also added text about non-linear SMOTE variants and how their geometric attacks might or might not extend to them.
- Reviewer `cZmi` agreed that the worst-case geometric counterexamples are measure-zero events and suggested explicitly phrasing results as holding “almost surely” or “with probability 1,” which we did. They were broadly satisfied with the edits and clarifications, and increased the score from 4 to 8 -- see note `qlWKjPwloJ`.
- Reviewer `Jfyn` also increased their score from 4 to 6 (see note `4iQh47YqaM`), understanding our clarifications and revisions re. proofs of Theorems 1-4.
- We primarily focused our interaction with reviewer `grKo` on their concerns about framing and relevance. We believe our rebuttal helped address those concerns, clarifying that we are not positioning SMOTE as a designed privacy tool, but documented its widespread use on sensitive data and as a de facto baseline in synthetic-data and privacy evaluations. Reviewer `grKo` mentioned the revised framing/motivation is “quite convincing.”
- We also justified assumptions like k ≥ 3 by pointing to SMOTE’s common defaults; we also showed that our attacks remain effective even for k=2. The reviewer also appreciated that our revision “demonstrated stronger robustness.” We were working on clarifying whether our attacks would fail under non-linear SMOTE variants when the discussion closed. We believe we would have managed to address these remaining concerns with our additional experiments showing that our attacks remain highly effective even when a small amount of noise is added to the SMOTE-generated data -- see note `oKERSf4Znb` and Appendix C in the latest draft of the paper.

---

### Meta-Review · Area_Chair_sNzP · 2026-01-06

**Summary:**

The paper studies the privacy vulnerability of SMOTE for augmenting the imbalanced data and generating the synthetic data. The reviewers have raised several concerns initially, while most of them have clearly addressed at rebuttal stage. Reviewers cZmi and Jfyn said they will increase the score. Reviewer grKo also think the rebuttal is convincing. Based on the discussion, I recommend accept.

**Reviewer Concerns:**

The reviewers raise the following main concerns.
1. The assumption looks idealized.
2. Experiments are relatively simple.
3. The novelty should be clarified.

Based on the discussion, I think the main concerns have been addressed. Reviewer grKo additionally raised the question that whether the attack would fully fail under non-linear SMOTE variants. I think the author also provide the convincing response.

**Reviewer Scores:**

Reviewers cZmi and Jfyn have agreed to raise their scores. It is possible that Reviewer grKo also will raise the score, since the authors have addressed the main concerns.

---

### Decision · Program_Chairs · 2026-01-26

Accept (Poster)